# Regional Sources of Airborne Ultrafine Particle Number and Mass Concentrations in California

Xin Yu[1], Melissa Venecek[2], Anikender Kumar[1], Jianlin Hu[3], Saffet Tanrikulu[4], Su-Tzai Soon[4], Cuong Tran[4], David Fairley[4], and Michael J. Kleeman[1]*

[1]Department of Civil and Environmental Engineering, University of California, Davis. One Shields Avenue, Davis CA. [2]Department of Land, Air, and Water Resources, University of California, Davis. One Shields Avenue, Davis CA. [3]School of Environmental Science and Engineering, Nanjing University of Information Science and Technology. [4] Bay Area Air Quality Management District, San Francisco, CA.

*Corresponding author. Tel.: +1 530 752 8386; fax; +1 530 752 7872. E-mail address: mjkleeman@ucdavis.edu (M.J. Kleeman).

## Abstract

Regional concentrations and source contributions are calculated for airborne particle number concentration ($N_x$) and ultrafine particle mass concentration ($PM_{0.1}$) in the San Francisco Bay Area (SFBA) and the South Coast Air Basin (SoCAB) surrounding Los Angeles with 4 km spatial resolution and daily time resolution for selected months in the years 2012, 2015, and 2016. Performance statistics for daily predictions of $N_{10}$ concentrations meet the goals typically used for modeling of $PM_{2.5}$ (MFB< ± 0.5 and MFE < 0.75). The relative ranking and concentration range of source contributions to $PM_{0.1}$ predicted by regional calculations agree with results from receptor-based studies that use molecular markers for source apportionment at four locations in California. Different sources dominated regional concentrations of $N_{10}$ and $PM_{0.1}$ because of the different emitted particle size distributions and different choices for heating fuels. Nucleation (24-57%) made the largest single contribution to $N_{10}$ concentrations at the ten regional monitoring locations, followed by natural gas combustion (28-45%), aircraft (2-10%), mobile sources (1-5%), food cooking (1-2%), and wood smoke (0-1%). In contrast, natural gas combustion (22-52%) was the largest source of $PM_{0.1}$ followed by mobile sources (15-42%), food cooking (4%-14%), wood combustion (1-12%), and aircraft (2-6%). The study region encompassed in this project is home to more than 25M residents, which should provide sufficient power for future epidemiological studies on the

health effects of airborne ultrafine particles. All of the $PM_{0.1}$ and $N_{10}$ outdoor exposure fields produced in the current study are available free of charge at http://webwolf.engr.ucdavis.edu/data/soa_v2/monthly_avg2.

### 1. Introduction

Numerous epidemiological studies have identified positive correlations between exposure to ambient particulate matter (PM) and increased risk of respiratory and cardiovascular diseases, premature mortality and hospitalization (Pope et al., 2002;Pope et al.,

2004;Pope et al., 2009;Dockery and Stone, 2007;Ostro et al., 2015;Ostro et al., 2006;Ostro et al., 2010;Brunekreef and Forsberg, 2005;Fann et al., 2012;Gauderman et al., 2015;Miller et al., 2007).  Most of these studies have not fully addressed ultrafine particles (UFPs; Dp<0.1µm) because these particles make a very small contribution to total ambient PM mass (Ogulei et al., 2007). Toxicity studies suggest that UFPs may be

especially dangerous to human health since they have higher toxicity per unit mass (Li et al., 2003;Nel et al., 2006;Oberdorster et al., 2002) and can penatrate the lungs and enter the bloodstream and secondary organs (Sioutas et al., 2005). These toxicology results are suggestive but more epidemiological evidence is required before the threat to public health from UFPs can be fully assessed.

Most previous UFP epidemiology studies are based on particle number concentration ($N_x$ – the number of particles with diameter less than X nm) measured at fixed sites using commercially-available instruments.  These devices are expensive and they require regular maintence which limits the number of measurement sites that can be deployed. Translating measured $N_x$ into population exposure estimates is also difficult because UFP

concentrations change more rapidly over shorter distances than $PM_{2.5}$ (Hu et al., 2014b;Hu et al., 2015;Hu et al., 2014a).  Land use regression (LUR) models could potentially be used to interpolate UFP concentrations between sparse measurement locations, but the atmospheric processes governing $N_x$ concentrations are highly non-linear and (so far) sufficient training data is not generally available for LUR models to

estimate $N_x$ exposure over a large enough population to support a definitive epidemiology

study (Montagne et al., 2015). Previous attempts to use regional reactive chemical transport models to predict $N_x$ in highly populated regions have focused on nucleation, yieldeding a wide range of predicted concentrations and only modest agreement with measurements when different nucleation algorithms were used (Elleman and Covert,

2009b;Zhang et al., 2010;Elleman and Covert, 2009a). Obtaining accurate exposure estimates to $N_x$ in highly populated regions therefore remains a major challenge in UFP epidemiological studies.

Recent work has examined UFP mass ($PM_{0.1}$) as an alternative metric for UFP exposure, and demonstrated that $PM_{0.1}$ can be predicted with reasonable accuracy over large

populations using regional reactive chemical transport models (Hu et al., 2014b;Hu et al., 2014a). The $PM_{0.1}$ exposure fields developed using this technique have been used in multiple epidemiological studies that revealed associations with mortality and pre-term birth (Ostro et al., 2015;Laurent et al., 2016). Despite the success of studies using $PM_{0.1}$, techniques that estimate $N_x$ exposure are still needed because a large number of ongoing

UFP studies are based on $N_x$ and it is possible that $PM_{0.1}$ and $N_x$ are associated with different types of health effects.

Here we extend the previous work using regional reactive chemical transport models for UFPs to include $N_x$ in the San Francisco Bay Area (SFBA) and the South Coast Air Basin (SoCAB) region around Los Angels which are the two most densely populated major

metropolitan location in California. Source contributions to $PM_{0.1}$ and $N_x$ are tracked using the University of California, Davis / California Institute of Technology (UCD/CIT) regional reactive chemical transport model with 4 km spatial resolution. Predicted concentrations during the year 2012 are compared to measurements available at ten regional monitoring sites. The spatial distribution fields of different particle metrics ($N_x$,

$PM_{0.1}$, $PM_{2.5}$) are combined with population distributions to estimate exposure. To the best of our knowledge, this is the first integrated study of both UFP number and mass using a regional reactive chemical transport model in California.

## 2. Model Description

The UCD/CIT chemical transport model used in the current study has been succesfully applied in sevaral previous studies in the San Joaquin Valley (SJV) and the SoCAB (Ying et al., 2008b;Ying et al., 2008a;Hu et al., 2015;Hu et al., 2017;Chen et al., 2010;Held et al., 2004;Held et al., 2005;Hixson et al., 2010;Hixson et al., 2012;Hu et al., 2012;Kleeman and Cass, 2001;Kleeman et al., 2007;Kleeman et al., 1997;Mahmud, 2010;Mysliwiec and Kleeman, 2002;Rasmussen et al., 2013;Ying and Kleeman, 2006;Zhang and Ying, 2010). The model includes algorithms for emissions, transport, dry deposition, wet deposition, gas phase chemistry, gas-to-particle conversion, coagulation, and some condensed phase chemical reactions. Nucleation was added to the model for the first time in the current study using the ternary nucleation (TN) mechanism involving $H_2SO_4$-$H_2O$-ammonia ($NH_3$) (Napari et al., 2002). As was the case in previous studies using this algorithm, the resulting nucleation rate was adjusted using a tunable nucleation parameter set to $10^{-5}$ for new particle nucleation (Jung et al., 2010). The Kerminen and Kulmala (2002) parameterization was added in order to bridge the gap between the 1 nm particle nuclei and their appearance into the smallest size bin of the UCD/CIT model (~10 nm). The nuclei growth rate (GR) in the Kerminen and Kulmala (2002) parameterization is one of the factors that accounts for the competition between the condensation and nucleation of over-saturated compounds until the nucleated particles grow to the size of the smallest bin in the regional model, at which point this competition is represented explicitly by the model operators. In the current study, the GR for nucleated sulfate particles was calculated using the diffusion-limited condensation rate of sulfuric acid based on the recommendation of Kerminen and Kulmala. Once particles reach ~10nm, the full operators in the model calculations predict growth by condensation of sulfuric acid, nitric acid, ammonia, and secondary organic aerosol (SOA). Perturbation studies were conducted in the current analysis to test the effect of GR with a box model configured to represent a single grid cell using the full set of model operators. Initial conditions in the SAPRC11 gas-phase mechanism were 0.04 ppm $O_3$, 0.05 ppm NO, 0.0 ppm $NO_2$, 0.05 ppm HCHO, 0.1 ppm ISOPRENE, 0.1 ppm BENZENE, and 0.01 ppm ALK5. A nucleation event was initiated at 8am by setting $H_2SO_4$ concentrations to $10^7$ molecules $cm^{-3}$ and $NH_3$ concentrations to 100 ppt. The

nominal GR was multiplied by a factor ranging from 0.5 to 2.0 to test the sensitivity of

120    the model results. Figure 1 illustrates the growth of nucleated particles between 5am and

12 noon for conditions representing July in California. The number concentration of

nucleated particles increases from zero to values between 2500 - 3000 # cm$^{-3}$.  SOA

condenses on the particles causing their size to increase above 100nm. Coagulation and

deposition processes remove particles over time.  Three separate simulations are

illustrated in Figure 1 using the nominal GR along with perturbations of 0.5*GR and

2.0*GR.  These model perturbations fall almost exactly on top of the basecase

simulations, suggesting that results are not overly sensitive to GR during the first few

seconds of nuclei growth before calculations are handed off to the regional model

algorithms.

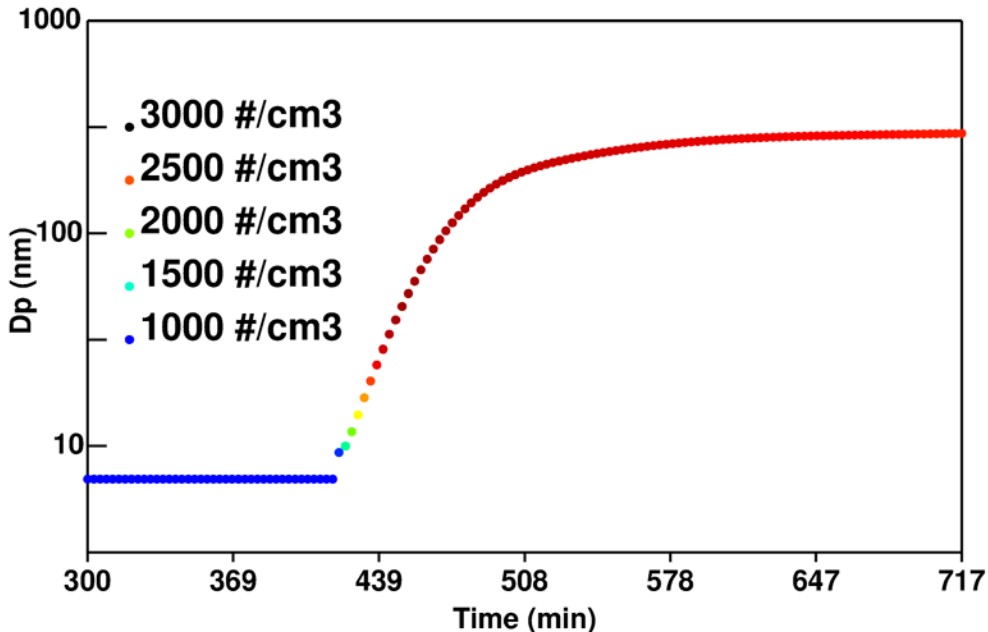


Figure 1: Simulated particle nucleation event followed by growth due to SOA

condensation under conditions representing July in California.  Vertical axis displays the

mean diameter of the nuclei mode while color represents the particle number

concentration.

Several previous modeling studies have been conducted to evaluate the performance of

the ternary nucleation mechanism on predicted $N_x$ using global and regional models.

Jung et al. (2010) found that a scaled version of the ternary $H_2SO_4$-$NH_3$-$H_2O$ nucleation

theory (Napari et al.,2002 with a supplemental $10^{-5}$ nucleation tuning factor) added to the PMCAMx-UF model produced $N_x$ predictions in reasonable agreement with

observations. The study of Westervelt et al. (2013) also showed that the ternary nucleation parameterization (with a supplemental $10^{-5}$ nucleation tuning factor) added to the Goddard Earth Observing System global chemical transport model (GEOS-Chem) produced reasonable $N_x$ predictions on average when compared with measurements at five locations spanning various environments.  Jung et al. (2008) considered multiple

nucleation parameterizations in the Dynamic Model for Aerosol Nucleation (DMAN)  to predict the nucleation events and non-events observed during the Pittsburgh Pittsburgh Air Quality Study (PAQS) conducted between July 2001 and September 2002. Their results showed that the ternary nucleation mechanism ((Napari et al., 2002) with a supplemental $10^{-5}$ nucleation tuning factor) was a suitable nucleation scheme for 3-D

chemical transport models. Although there have been numerous significant efforts to incorporate nucleation algorithms into three-dimensional regional and global models (Jung et al., 2008;Jung et al., 2010;Westervelt et al., 2013;Zhang et al., 2010;Yu et al., 2015;Dunne et al., 2016;Fanourgakis et al., 2019), nucleation modeling studies are still in the early stages of development and further efforts are needed to reduce the uncertainty in

both the nucleation rate and growth mechanisms.

In the current study, emission, transport, deposition, and coagulation of UFPs were simulated using operators developed for the UCD/CIT model framework, leading to modification of the particle size distribution and the subsequent $N_x$ concentrations. Dynamic condensation / evaporation is considered for all particle size bins with predicted

UFP growth rates of 2-3 nm $hr^{-1}$ or higher under favorable conditions.  The regional model operators are not well suited for the most extreme changes to the particle size distribution that occur within the first few seconds or minutes after emissions to the atmosphere (such as within 300 m of roadways).  Dedicated simulations can predict the dynamic condensation/evaporation of particles at distances of 10's of meters downwind

of the roadway (Zhang et al., 2005;Zhang et al., 2004) mostly due to the partitioning of SOA (Anttila and Kerminen, 2003;Trostl et al., 2016), but these calculations are too expensive for domains spanning thousands of km.  Regional calculations such as those illustrated in the current study rely on emissions characterization measurements that

include a few minutes of aging to capture the "near-field" emissions of particle size and

composition that can then be used as the starting point for regional model calculations. In some cases, evaporation of UFPs in the first few seconds after release to the atmosphere is therefore represented by reducing the primary emissions of nano-particles based on measurements conducted at high dilution factors (Xue et al., 2018a) or using measurements of particle volatility to estimate the evaporation at high dilution factors

(May et al., 2013a;May et al., 2013b;Kuwayama et al., 2015). All of the results presented in the current analysis focus on regional UFP concentrations with 4km resolution.

The model domains used in the study are shown in Figure 2. The parent domain with 24 km horizontal resolution covered the entire state of California (referred to as CA_24 km) and the two nested domains with 4 km horizontal resolution covered the SFBA + SJV +

South Sacramento Valley air basins (referred as SJV_4 km) and the SoCAB surrounding Los Angeles (referred as SoCAB_4 km). The UCD/CIT model was configured with 16 vertical layers up to a height of 5 km above ground level, with 10 layers in the first 1 km. Previous studies have shown that this vertical configuration captures the air pollution system above California (Hu et al., 2014a;Hu et al., 2014b;Hu et al., 2015). Particulate

number, mass, and composition are represented in 15 size bins, with particle diameters being centered within equally spaced logarithmic size interval spanning the diameter range from 0.01 to 10µm. Nucleated particles were initialized in a 16th size bin with initial diameter of 0.007 µm.

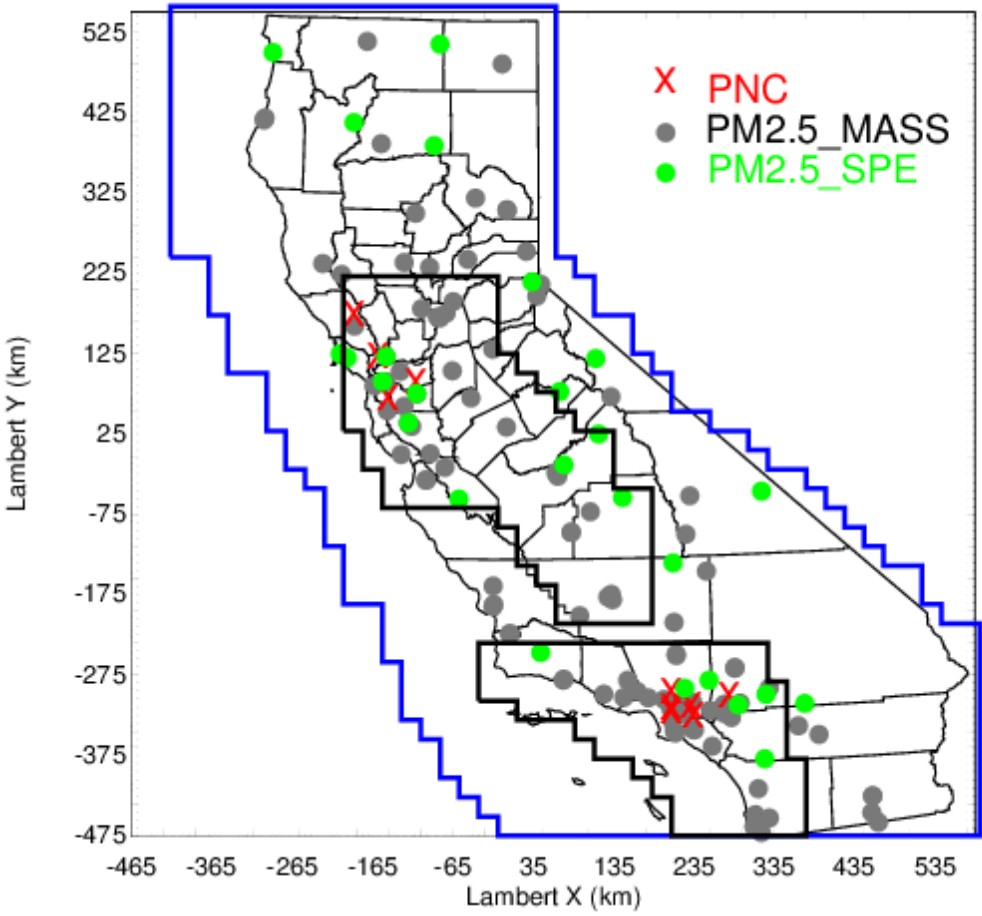


Figure 2: Modeling domains. Blue lines outline the CA_24 km domain, black lines outline the SoCAB_4 km (bottom) and SJV_4 km domains (top). Red crosses represent ten $N_x$ sites (four sites operated by staff at the Bay Area Air Quality Management District (BAAQMD) and six sites from the Multiple Air Toxics Exposure Study IV (MATES

IV)). Detailed location information for the $N_x$ sites is listed in Table S3. Green dots represent BAAQMD PM$_{2.5}$ speciation network sites and the Interagency Monitoring of Protected Visual Environments (IMPROVE) sites; gray dots represent the PM$_{2.5}$ federal reference method (FRM) sites.

### 2.1 Meteorological Fields

Hourly meteorological fields during the modeling period were generated by the Weather Research and Forecasting (WRF) model version 3.4 with three nested domains that had horizontal resolutions of 36 km, 12 km and 4 km, respectively. In the present simulations, the WRF model was configured with 50 vertical layers (up to 100 hpa) and four-

dimensional data assimilation (FDDA) nudging was utilized to improve the agreement

between model predictions and observed meteorological patterns (Otte, 2008b, a). WRF predictions for wind speed, temperature, and relative humidity were compared to measurements for seven counties in the SFBA and two counties in SoCAB (see Table S2). Temperature has mean bias (MB) within ~0.2 °C and root- mean-square errors (RMSE) between 4-5 °C. Wind speed has mean fraction bias (MFB) within ±0.20 and

RMSE generally <2.0 m/s. This level of performance is consistent with performance of WRF in previous studies conducted in California (Zhao et al., 2011;Hu et al., 2015).

### 2.2 Emissions

The emission inventories used in the SFBA were developed by the BAAQMD for the year 2012 based on the regulatory inventory provided by the California Air Resources

Board for that same year. The SFBA inventory was processed using the Sparse Matrix Operator Kernel Emissions (SMOKE) v3.7 software package provided by US EPA. SMOKE was configured to separately tag emissions from on-road gasoline vehicles, off-road gasoline vehicles, on-road diesel vehicles, off-road diesel vehicles, food cooking, biomass burning, non-residential natural gas, and all other sources. The emission

inventories used in South Sacramento Valley, SJV and SoCAB were provided by the California Air Resources Board.

Measurements conducted in parallel with the current study found that particles emitted from natural gas combustion in home appliances were semi-volatile when diluted by a factor of 25 in clean air, but particles emitted from reciprocating engines did not

evaporate under the same conditions (Xue et al., 2018a). Near-field emissions from all natural gas sources combustion sources other than reciprocating engines were therefore set to 30% of their nominal levels. A map of the natural gas emissions distribution is shown in Supporting Information (Figure S3).

SMOKE results were transformed into size-resolved emissions of particle number, mass, and composition using measured source profiles through an updated version of the emissions model described by Kleeman and Cass (1998). The PM profiles used for each

source type were specified as weighted averages from each of the detailed sources within each broad category as summarized in Table S1. Detailed PM source profiles for major sources of ultrafine particulate matter are based on measurements conducted during source tests (Li and Hopke, 1993;Kleeman et al., 1999, 2000;Robert et al., 2007a;Robert et al., 2007b;Mazaheri et al., 2009). In most cases, these emissions size distributions strongly influence the size distributions of particles in the ambient atmosphere (see Figures S1 and S4).  A more detailed discussion of the emissions processing has been presented in a previous study (Hu et al., 2015).

## 3. Results

### 3.1 Statistical Evaluation

According to Taylor's Hypothesis (Shet et al., 2017), it is expected that the spatial distribution of model results is more important than the temporal distribution when evaluating performance. In the current study model performance evaluations are limited to the locations where measurements were made.  Therefore, the temporal distribution is also considered by comparing predicted vs. measured daily average $N_x$, $PM_{2.5}$ and individual $PM_{2.5}$ species mass concentrations.

The evaluation data set was compiled from several measurement networks including the sites operated by staff at the Bay Area Air Quality Management District (BAAQMD), the IMPROVE sites, the MATES IV sites and FRM sites. In order to account for the uncertainty in predicted wind fields and spatial surrogates used to place emissions, "best-fit" model results were created by identifying the closest match within 3 grid cells of each measurement location.  "Best-fit" model performance for $PM_{2.5}$ at routine monitoring sites (Figure 2) meets the performance criteria suggest by Boylan and Russell (Boylan and Russell, 2006) (mean fractional error (MFE) $\leq$ +0.75 and mean fractional bias (MFB) $\leq$ $\pm0.5$) (Table S4).  Table S5 shows the MFB and MFE values of gaseous species of $O_3$, NO, $NO_2$, CO and $SO_2$ using daily averages across all measurement sites during the entire simulated period. Gaseous species of $O_3$, CO, NO, $NO_2$ and $SO_2$ have MFBs within $\pm$ 0.3 and MFE less than 0.5, indicating consistent behavior between predictions and measurement for these species. The ability of UCD/CIT predictions for key gas species,

mass and chemical component concentrations in the $PM_{0.1}$ and $PM_{2.5}$ size fractions was also evaluated in previous studies (Ying and Kleeman, 2006;Ying et al., 2008a;Ying et al., 2008b;Hu et al., 2012;Chen et al., 2010;Held et al., 2005;Hu et al., 2015;Hu et al.,

2017;Venecek et al., 2018). The performance of the UCD/CIT air quality model in these studies generally meets standard model performance criteria. Of greatest interest in the current study, predicted "best-fit" $N_{10}$ values were compared to measured $N_7$ values at four sites in the SFBA (Santa Rosa, San Pablo, Redwood City and Livermore) and six sites in SoCAB (Anaheim, Central Los Angeles, Compton, Huntington, Inland-valley and

Rubidoux). $N_7$ measurements in the SFBA were made using an Environmental Particle Counter (EPC) Monitor Model 3783 (TSI Inc) while $N_7$ measurements in the SoCAB were made with EPC Model 3781 (TSI Inc).  Both monitors can detect ultrafine particles down to 7 nm which is smaller than the first size bin of 10 nm used in model calculations. Previous studies conducted at Fresno, California, suggest that $N_{7-10}$ accounts for

approximately 8% of $N_7$ (Watson et al., 2011), and so some amount of negative bias is expected when comparing predicted $N_{10}$ to measured $N_7$.  The evaluation results for "best-fit" $N_{10}$ summarized in Table 1 follow this expected trend but mean fractional bias (MFB) and mean fractional error (MFE) at each comparison site still meet the $PM_{2.5}$ performance criteria suggested by Boylan and Russell (2006). This level of performance

is comparable to the results from a previous UFP number simulation conducted in Northern California using a modified version of the WRF-Chem model (Lupascu et al., 2015).   The level of agreement between predicted "best-fit" and measured $PM_{2.5}$, individual $PM_{2.5}$ species, key gas species and $N_{10}$ builds confidence in the model skill for UFP predictions in the current study.


Table 1. Performance statistics for "best-fit" $N_{10}$ predictions vs. $N_7$ at individual monitoring sites. Threshold for PM modeling applications is typically MFB< ± 0.5 and MFE < 0.75.

| | Ave Obs. Particles cm$^{-3}$ | Ave Sim. Particles cm$^{-3}$ | R | MFB | MFE | RMSE Particles cm$^{-3}$ |
|---|---|---|---|---|---|---|
| Livermore | 8219 | 9201 | 0.31 | 0.10 | 0.09 | 3615 |
| Redwood city | 11500 | 11325 | 0.97 | 0.02 | 0.08 | 1132 |
| San Pablo | 10481 | 15822 | 0.45 | 0.30 | 0.31 | 10302 |
| Santa Rosa | 8655 | 8967 | 0.78 | 0.05 | 0.15 | 2063 |
| Anaheim | 12850 | 14812 | 0.74 | 0.12 | 0.14 | 4239 |
| Central LA | 17378 | 25376 | 0.31 | 0.37 | 0.38 | 10328 |
| Compton | 16203 | 21036 | 0.36 | 0.24 | 0.26 | 8127 |
| Huntington | 23207 | 24103 | 0.77 | 0.04 | 0.08 | 3698 |
| Inland-Valley | 15028 | 16875 | 0.37 | 0.12 | 0.17 | 4290 |
| Rubidoux | 10728 | 11920 | 0.66 | 0.11 | 0.16 | 3069 |

Table 2 below summarizes the predicted correlations between daily-average particle number concentrations and PM$_{2.5}$ along with the measured correlations for these metrics. Measured correlations (R) are less than 0.5 at all locations except Santa Rosa where correlations are above 0.75. Model predictions for daily-average particle number concentrations and PM$_{2.5}$ are more highly correlated, with R ranging from 0.47 to 0.85.

The higher correlation between particle number vs. PM$_{2.5}$ in the predicted concentrations suggests that the model does not capture all of the complexity in the real atmosphere. Locations with high R values such as central Los Angeles also have the highest MFB and MFE and so the high correlation between particle number and PM$_{2.5}$ may reflect inaccuracies in the model inputs. At other locations where traditional model performance

metrics suggest that predictions are more accurate, the high correlation between particle number and PM$_{2.5}$ may be related to the model grid resolution. The 4km grid resolution used in the calculations smooths the sharp spatial gradients in the ultrafine particle concentration fields (see Figure 4 below). This same issue makes it difficult for point source measurements to accurately represent 4km average number concentrations. The

particle number concentrations measured at a fixed monitoring location may not represent the variation in particle number concentrations a few km away. PM$_{2.5}$

concentration gradients are smoother, making model predictions and point measurements easier to compare.

Table 2. Daily-average correlation ($R^2$) between PM2.5 mass and particle number concentration at 8 sites in California.

| R | Livermore | Redwood City | San Pablo | Santa Rosa | Anaheim | Central LA | Compton | Rubidoux |
|---|---|---|---|---|---|---|---|---|
| Obs | 0.20 | 0.10 | 0.40 | 0.76 | 0.28 | 0.37 | 0.39 | 0.47 |
| Sim | 0.53 | 0.70 | 0.74 | 0.47 | 0.71 | 0.85 | 0.78 | 0.71 |

### 3.2 $PM_{0.1}$ and $N_{10}$ Source Apportionment in California

The UCD/CIT model uses a moving sectional approach to conserve particle number and mass while letting particle radius change due to condensation and evaporation (Kleeman et al., 1997). The method to calculate source contributions to number concentration is performed for each moving section individually. Number is explicitly conserved and correctly apportioned to sources in this algorithm. Each particle source type / moving size bin includes an artificial tracer equal to 1% of the primary particle mass. The mass of this tracer is related to the number of particles by the equation

$$\text{tracer\_source\_i} * 100 = \text{N\_source\_i} * 3.14159/6 * \text{Dp\_bin} * \rho\_i \quad (eq1)$$

where $\rho\_i$ is the density of primary particles emitted from source i. This equation can be easily rearranged to solve for N_source_i as a function of tracer_source_i in each size bin. Condensation/evaporation changes the particle diameter as semi-volatile components move on and off the particle but this does not change tracer_source_i or N_source_i. As a result, the moving sectional approach greatly simplifies the source apportionment of particle number compared to other models that use fixed particle size bins with condensation / evaporation transferring material between bins.

Coagulation complicates source apportionment calculations for particle number because coagulation events conserve particle mass but destroy particle number. The model calculations treat the most frequently occurring coagulation events between very small particles and very large particles in a manner analogous to condensation. When two

particles coagulate, the mass of the smaller particle is added to the mass of the larger

particle.  The number concentration of the smaller particle is discarded while the number

concentration of the larger particle stays constant. This slightly reduces the accuracy of

source apportionment calculations for particle number in the larger size bins because the

tracer_source mass in the larger size bin is no longer proportional to the number

concentration from that source.  This issue is relatively minor since size bins larger than

1µm that act as the dominant sink during particle coagulation events typically account for

less than 5% of the total number concentration.

Perturbation studies were conducted to test the accuracy of the source apportionment

calculations by setting the UFP emissions for on-road gasoline vehicles to zero during the

month August 2012.  Emissions of gases and emissions of larger particles from on-road

vehicles were not changed.  The difference between this perturbation simulation vs. the

basecase simulation was calculated to estimate the number concentration of particles

associated with on-road gasoline vehicles.  This "zero-out" concentration was then

compared to the standard model source-apportionment calculations in Figure 3 below.

The two methods for number source apportionment yield very similar spatial patterns and

very similar maximum concentrations of ~0.5 kcounts cm$^{-3}$.  The tracer source

apportionment method accounts for all particle sizes, which produces slightly higher

concentrations than the zero-out method that only considered particles smaller than 100

nm.

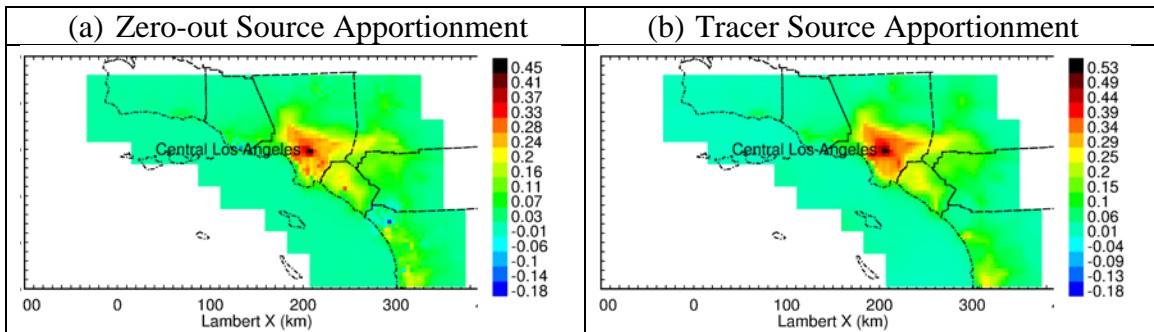

Figure 3: Particle number concentrations associated with on-road gasoline vehicles
calculated using the zero-out method and the artificial tracer method in August 2012.


Many of the spatial patterns measured for airborne particle number concentrations in past studies have focused on the gradients around roads (see for example (Zhu et al., 2002a;Zhu et al., 2002b;Zhang et al., 2005;Zhang et al., 2004;Sowlat et al., 2016).  These gradients are impossible to resolve using a regional model with 4km resolution.  A

limited set of additional simulations were conducted using the WRF/Chem model configured with Large Eddy Simulation (LES) around Oakland California so that spatial scales down to 250m could be examined.  Maps of the predicted ultrafine particle mass concentrations for gasoline, diesel, food cooking, wood combustion, and natural gas combustion particles are shown in Figure 4 below.  At 250m resolution, ultrafine

particles from diesel engines peak on major transportation corridors while ultrafine particles from gasoline vehicles are more diffuse reflecting their increased activity on adjacent surface streets.  Ultrafine particles from natural gas combustion are even more diffuse reflecting contributions from area sources across the region.  As the spatial resolution decreases to 1km and then 4km, the fine details around roadways are

artificially diluted in the larger grid cells.  This process shifts the dominant source of ultrafine particles over roadways from diesel engines at 250m resolution to natural gas combustion at 4km resolution.  These simulation results are consistent with measurements of particle number in the proximity of roadways which show that the traffic contribution to particle number concentration decays to background levels within

300 m (Zhu et al., 2002a;Zhu et al., 2002b). The measurements made by Zhu et al. indicate that the traffic contribution to regional number concentration cannot be distinguished from other sources on a regional scale using 4km grid cells which is the focus of this study.

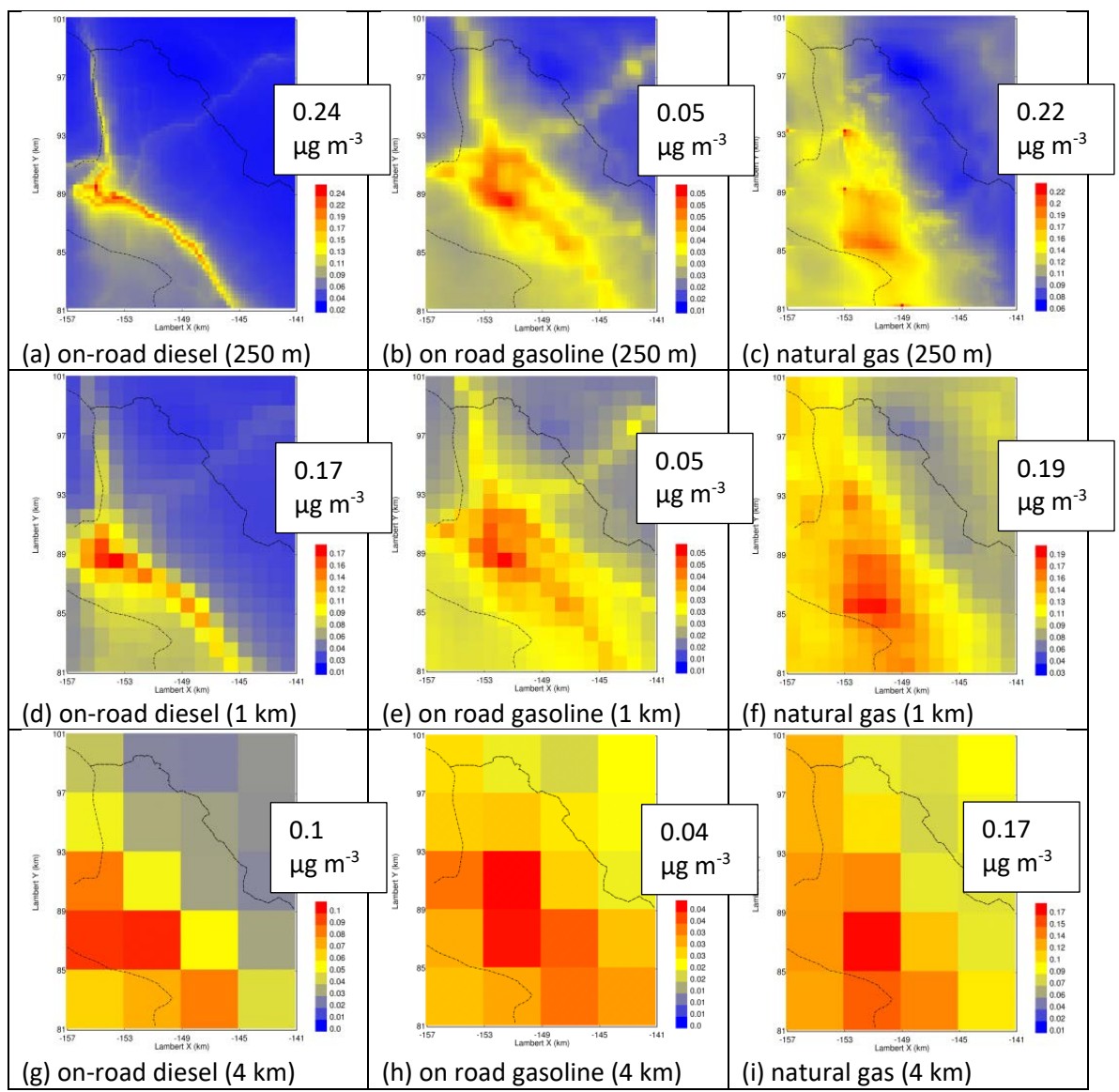

Figure 4: PM$_{0.1}$ mass concentration associated with on-road diesel, on-road gasoline, and natural gas combustion at 250m, 1km, and 4km resolution over Oakland, California.

**3.2.1 UCD/CIT PM$_{0.1}$ source contributions compared to Chemical Mass Balance (CMB) results**

A recently completed study measured the composition of PM$_{0.1}$ at four sites in California and calculated source contributions using molecular markers (Xue et al., 2018b). Figures 5 and 6 compare the source contributions to PM0.1 OC concentrations predicted by the

UCD/CIT model and "measured" using the molecular marker technique at San Pablo, East Oakland, downtown Los Angeles and Fresno during a summer month (August 2015) and a winter month (February 2016). The "others" category in the molecular marker calculation represents unresolved sources, while in the UCD/CIT model "others" represents the sum of non-residential natural gas source combustion, aircraft emissions,

and the sources that were not tagged in the current study. In general, the ranking and concentration range of source contributions to PM0.1 OC from the molecular marker technique and the UCD/CIT model are consistent. Natural gas dominates PM0.1 OC in the summer of 2015 at San Pablo, East Oakland, downtown Los Angeles and Fresno, while wood smoke and aircraft are the major sources of PM0.1 OC in Fresno and East

Oakland during the winter of 2016. The importance of ultrafine particles from natural gas combustion has not previously been recognized because these particles lack a unique chemical signature, which causes them to be lumped into the "unresolved" category in receptor-based source apportionment studies. The source contribution results for the gasoline, diesel, wood burning, meat cooking and other source categories predicted by

the UCD/CIT model and the molecular marker technique illustrated in Figures 5 and 6 build confidence in the accuracy of the UFP source predictions in the current study.

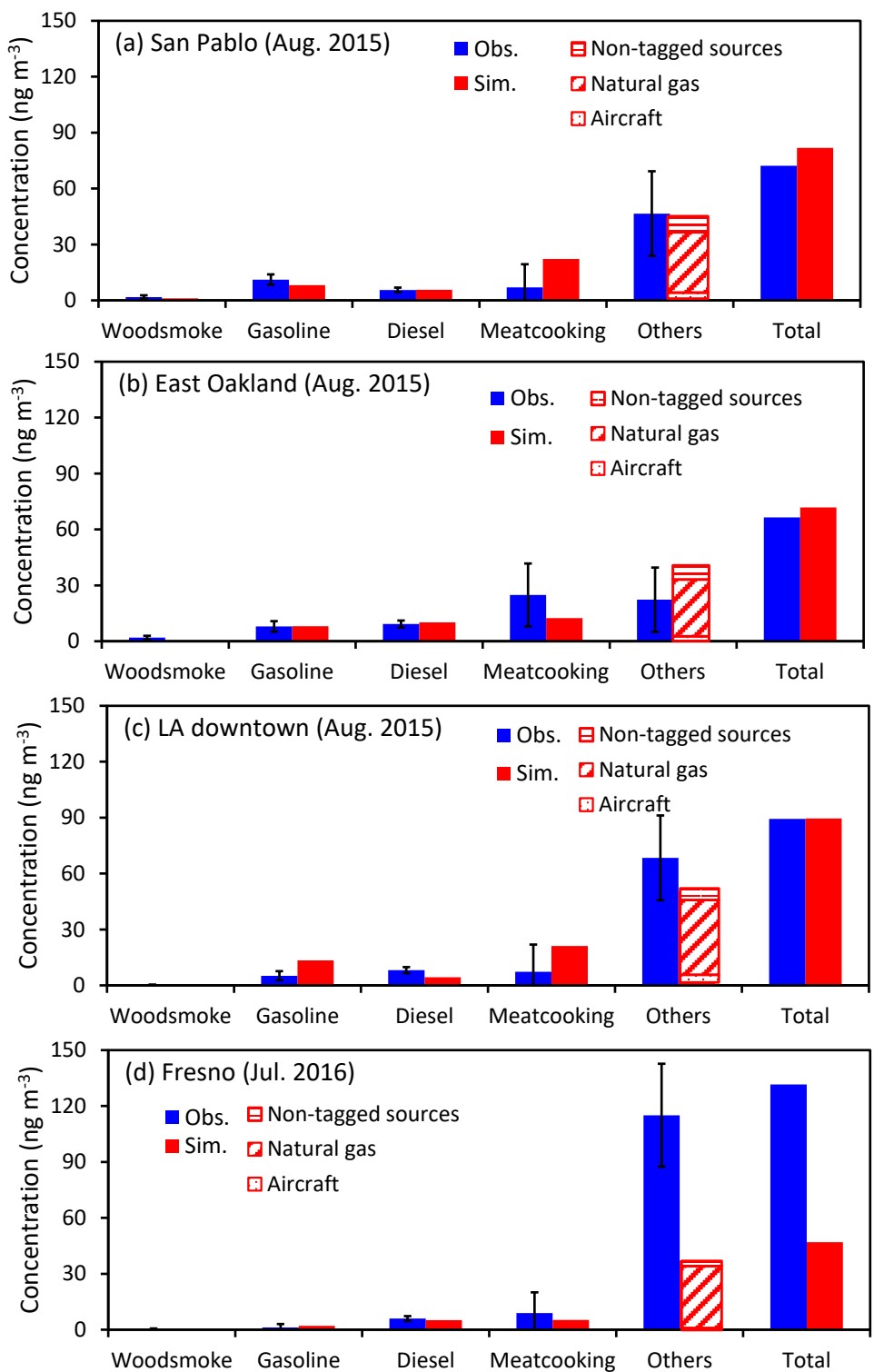

Figure 5: Source contribution to PM0.1 predicted by the CMB receptor model and the UCD/CIT model at four sites in California in August 2015. CMB results are calculated using 3-day average measurements composited for a full month.


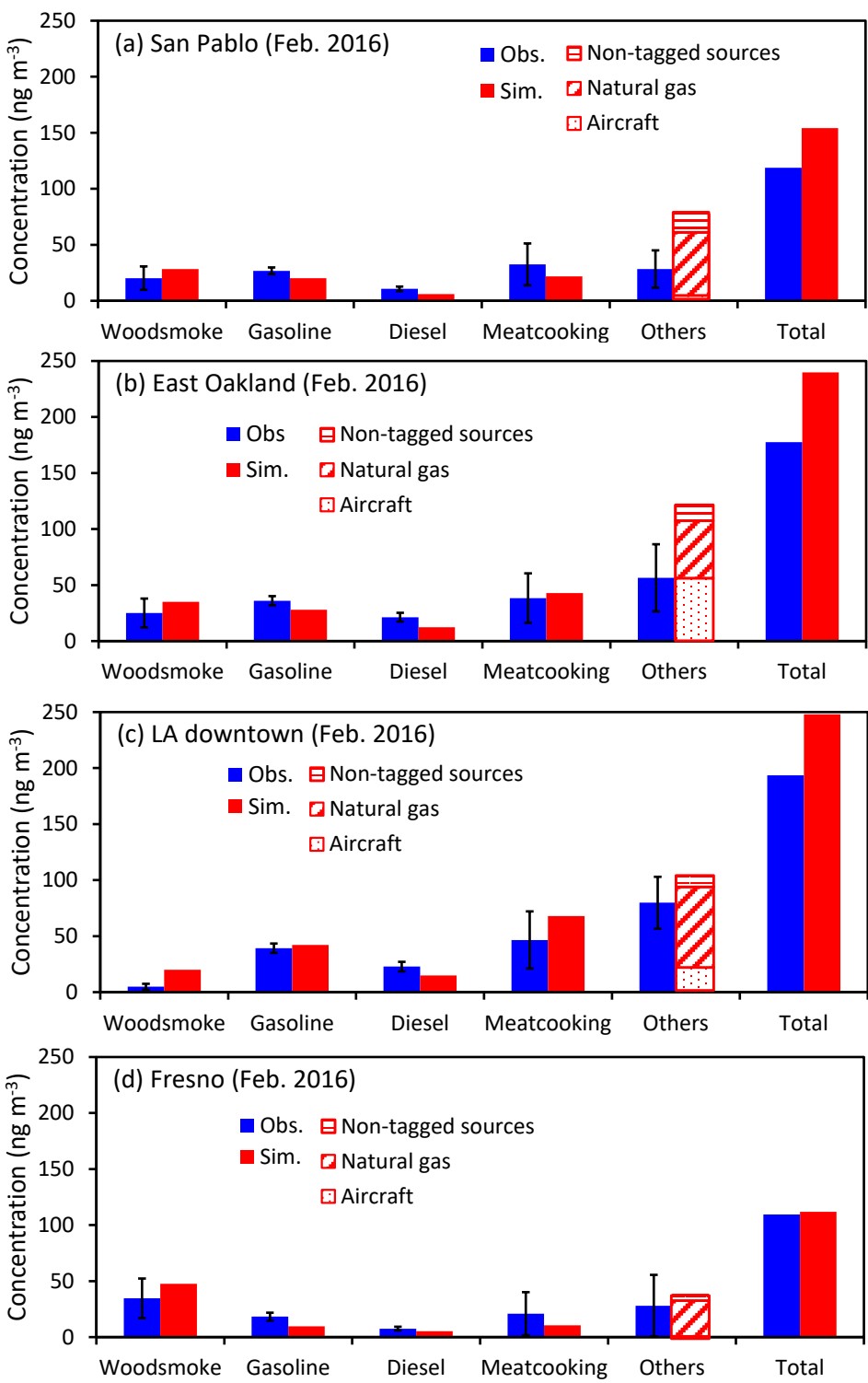

Figure 6: Source contribution to PM0.1 predicted by the CMB receptor model and the UCD/CIT model at four sites in California in February 2016. CMB results are calculated using 3-day average measurements composited for a full month.

**3.2.2 $PM_{0.1}$ and $N_{10-1000}$ Source contributions in California**

Figures 7-9 and 10-12 show the seasonal variation of major source contributions to primary $N_{10}$ and $PM_{0.1}$, respectively. The black circles in Figure 7-9 represent the measured $N_{7-1000}$ at four BAAQMD sites in SFBA and six MATES sites in Los Angeles and Riverside counties. Predicted "best-fit" $N_{10}$ follows the same trends as measured seasonal variations of $N_7$ at Livermore, Redwood City, Santa Rosa, Huntington Park, Inland Valley, and Rubidoux. The model over predicts $N_7$ at Anaheim, central Los Angeles, and Compton but overall model performance statistics for $N_7$ are within the target range for $PM_{2.5}$ applications (see Table 1). Nucleation contributes to $N_{10}$ at all sites but makes negligible contributions to $PM_{0.1}$ concentrations. Traffic sources including gasoline- and diesel-powered vehicles make significant contributions to $PM_{0.1}$ concentrations at each measurement site depending on proximity to major freeways. Near-roadway effects on ultrafine particle concentrations are not apparent since these locations were chosen to be regional monitors and so they are more than 300 m from the nearest freeway. Predicted contributions from traffic sources are consistent with the molecular marker results illustrated in Figures 5-6. Traffic contributions to regional $N_{10}$ concentrations more than 300 m away from roadways are even smaller than $PM_{0.1}$ contributions because the size distribution of particles emitted from motor vehicles peaks at 100 – 200 nm (Robert et al., 2007a;Robert et al., 2007b). Wood smoke makes strong contributions to regional $PM_{0.1}$ concentrations in central California during winter but much smaller contributions in the SoCAB because wood burning is not typically used for home heating in this region. Wood burning contributions to $N_{10}$ are less dominant in central California because the size distribution of particles emitted from wood combustion peaks at 100-300 nm (Kleeman et al., 2008b). The largest primary source of $N_{10}$ in central California and $N_{10}+PM_{0.1}$ in the SoCAB is natural gas combustion. Industrial processes and power generation that use natural gas do not follow strong seasonal cycles and so the strength of the natural gas source contributions is somewhat constant across seasons subject to variability caused by meteorological conditions.

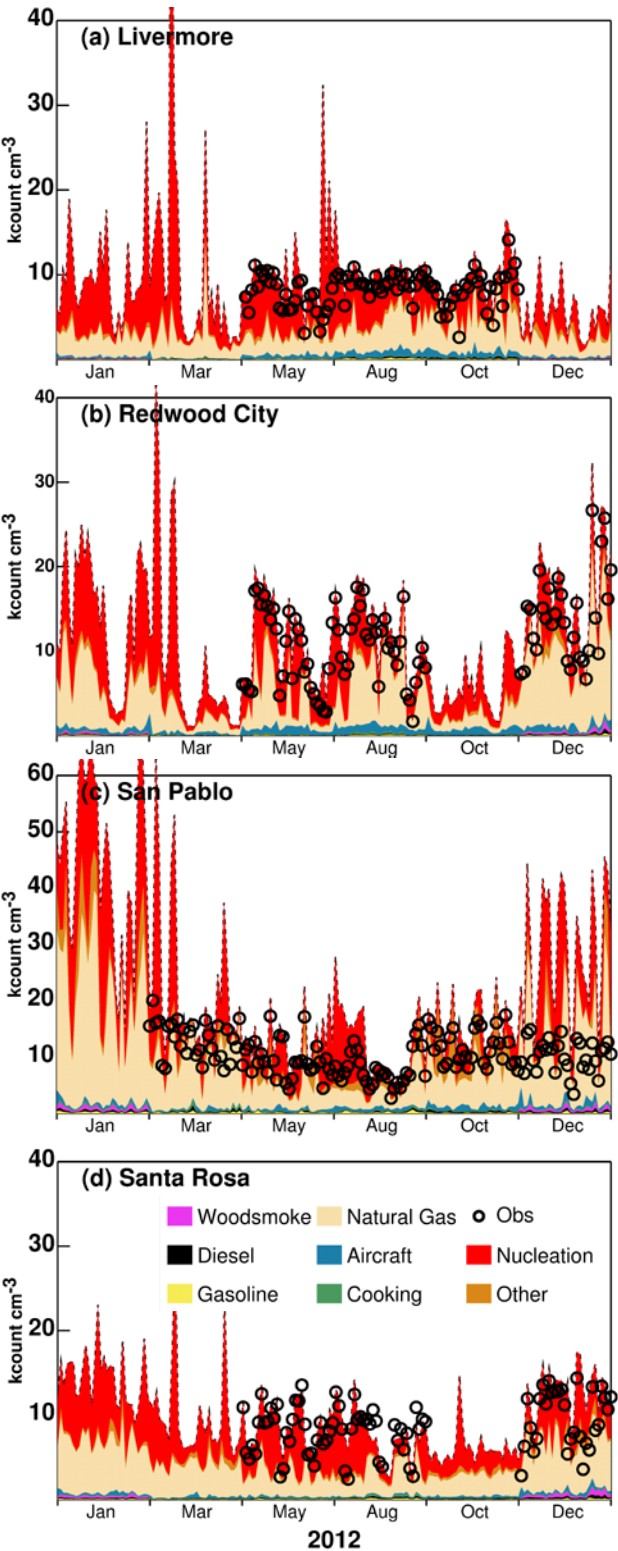

Figure 7: Seasonal variation of measured $N_7$ (black circles) and major source contributions to "best-fit" $N_{10}$ at Livermore, Redwood City, San Pablo and Santa Rosa, respectively. Results within each month have daily time resolution.

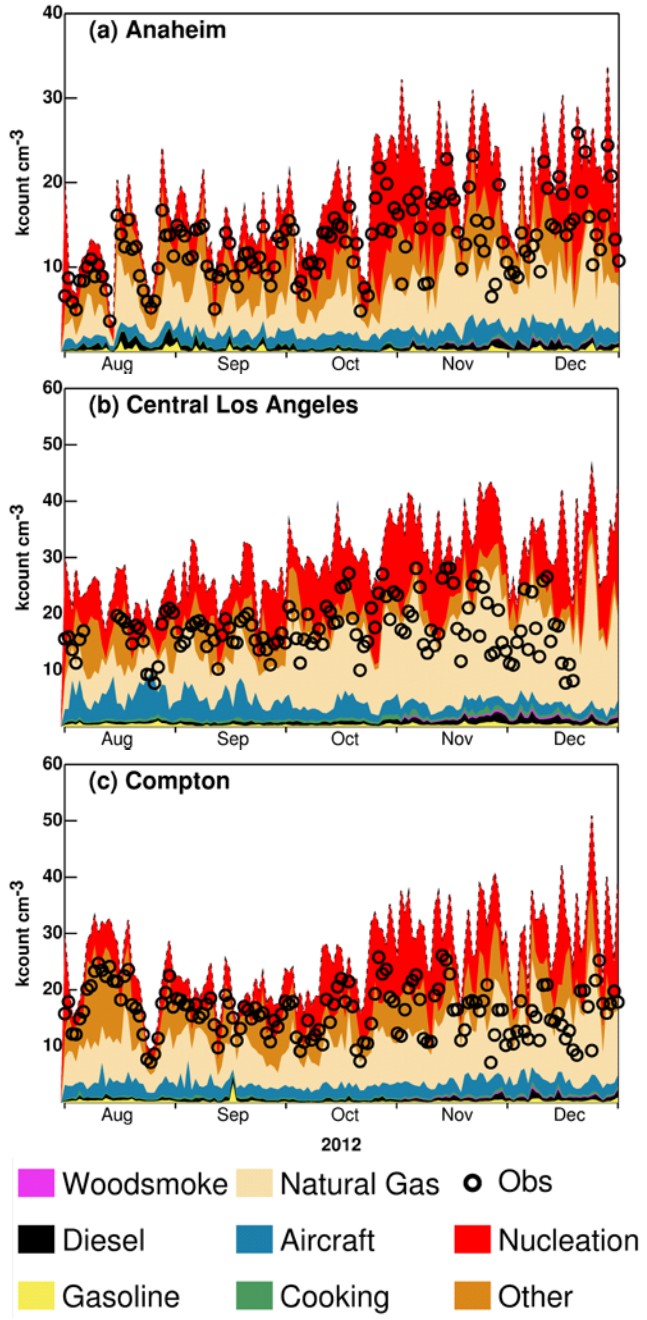

Figure 8: Seasonal variation of measured $N_7$ (black circles) and major source contributions to "best-fit" $N_{10}$ at Anaheim, Central LA, and Compton, respectively. Results within each month have daily time resolution.

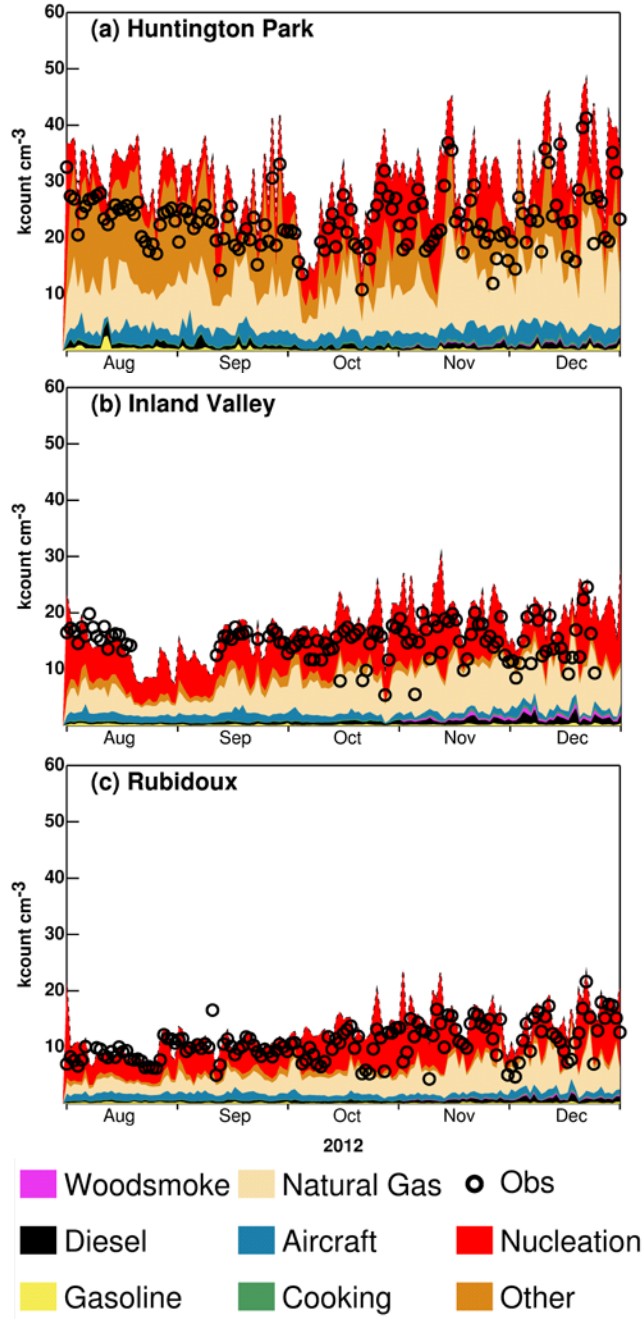

Figure 9: Seasonal variation of measured $N_7$ (black circles) and major source contributions to "best-fit" $N_{10}$ at Huntington, Inland-Valley, and Rubidoux, respectively. Results within each month have daily time resolution.

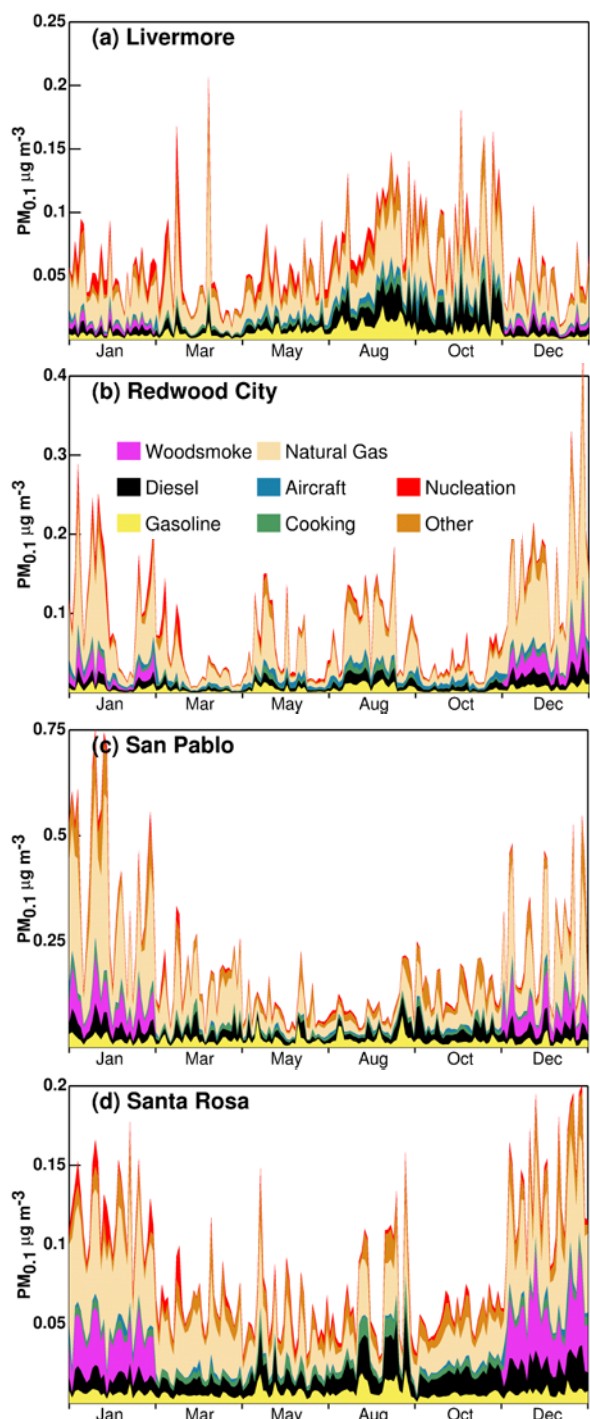

Figure 10: Seasonal variation of major source contributions to $PM_{0.1}$ at Livermore, Redwood City, San Pablo and Santa Rosa, respectively. Results within each month have daily time resolution.

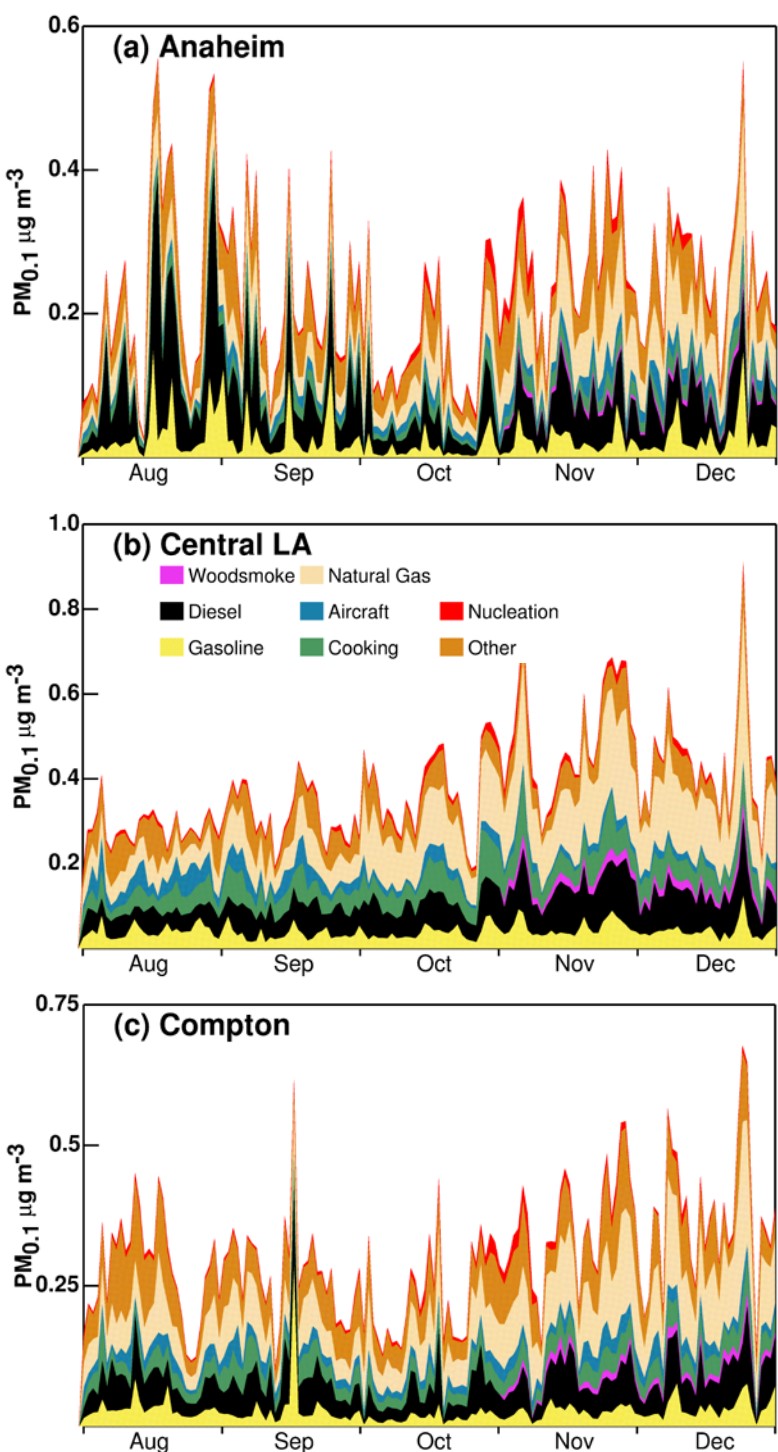

Figure 11: Seasonal variation of major source contributions to $PM_{0.1}$ at Anaheim, Central LA, and Compton, respectively. Results within each month have daily time resolution.


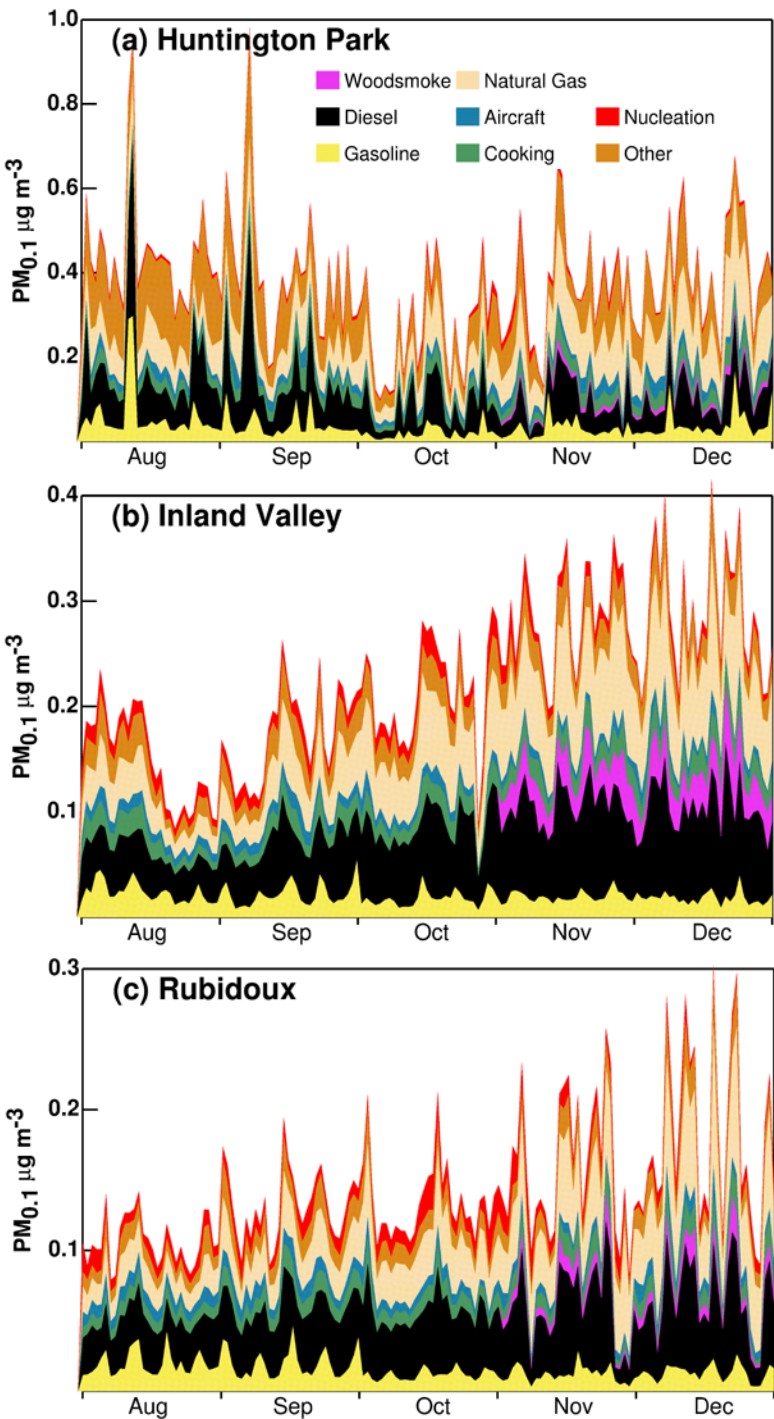

Figure 12: Seasonal variation of major source contributions to PM0.1 at Huntington, Inland-Valley, and Rubidoux, respectively. Results within each month have daily time resolution.

Figures 13 and Figures 14 show the source contributions to $N_{10}$ and $PM_{0.1}$, respectively, averaged over the days shown in Figures 7-9. Aside from nucleation, non-residential


natural gas combustion makes the largest predicted primary contribution to $N_{10}$ at all the sites that were evaluated. Traditional sources that were tracked including meat cooking, wood smoke, and mobile (gasoline + diesel) accounted for approximately 5-15% of the predicted $N_{10}$ at the sites selected for study. "Other" sources that were not tagged

explicitly in the current study accounted for 5-31% of $N_{10}$ across these sites. Nucleation is a significant source for of $N_{10}$ for both BAAQMD sites and MATES sites where sulfur emissions were highest, with contributions ranging from 24-57%.

The strong $N_{10}$ contribution from natural gas combustion reflects the emitted particle size distribution combined with the ubiquitous use of this fuel in the SFBA and SoCAB

regions. The chemical composition and size distribution information for non-residential natural gas combustion emissions used in this study was measured by Hildemann (1991) and Li and Hopke (1993), respectively. Size distributions and volatility were further confirmed during on-going field studies conducted by the current authors (Xue et al., 2018a). The estimated non-residential natural gas combustion particle number and mass

size distributions are shown in Figure S1 (left column). Clearly, the majority of particles from non-residential natural gas combustion are typically found in diameters <0.05 µm, while particles emitted from other sources such as wood combustion tend to have slightly larger particle diameter (with lower number concentration per unit of emitted mass). These natural gas particles grow through the condensation of SOA once in the

atmosphere, but they still contribute strongly to $N_{10}$ concentrations.

Figure 14 shows that on-road vehicles (gasoline and diesel combined) are the largest $PM_{0.1}$ source at Anaheim (39%), central LA (31%), Huntington Park (33%), Inland Valley (39%), and Rubidoux (42%), while natural gas combustion still makes the largest contribution to $PM_{0.1}$ at other evaluation sites. Contributions from cooking and mobile

sources are enhanced in $PM_{0.1}$ vs. $N_{10}$, with the cooking source accounting for 11% of $PM_{0.1}$ at Santa Rosa. The different rankings of source contributions to $N_{10}$ and $PM_{0.1}$ can be explained by the comparison of particle number-size distribution and particle mass-size distribution for the non-residential natural gas and wood burning sources at the four evaluated sites (Figure S1). Particles emitted from non-residential natural gas combustion

and wood burning have number distributions that peak at particle diameters of 0.016-

0.025 µm and 0.025-0.04 µm, respectively. Non-residential natural gas combustion and wood burning mass distributions, however, peak at particle diameters of 0.025-0.04 µm and 0.10-0.16 µm, respectively.

Figure 15-17 show diurnal variations of measured $N_{7-1000}$ and predicted "best-fit" $N_{10}$ averaged over days in August and December 2012. These months span the temperature range typically experienced across the year in California. Measured $N_{7-1000}$ diurnal patterns in August generally peak in the afternoon hours between 12-3pm with an optional morning peak around 6am. The main afternoon peak appears to be related to nucleation events while the smaller early-morning peak appears to be related to early morning human activity including natural gas combustion. The predicted "best-fit" $N_{10}$ diurnal variations in August followed the same trends as measurements at six out of ten sites (Livermore, Anaheim, Compton, Huntington Park, Inland Valley, and Rubidoux). The model failed to capture the mid-day nucleation event at Redwood City and Santa Rosa possibly due to missing $SO_2$ sources in the emissions inventory upwind from these sites. The model overestimated mid-day peak values at Anaheim and central Los Angeles. In December, the measured $N_{7-1000}$ diurnal pattern were more distinctly bimodal with the first peak around 7:00-8:00am and the second peak in the evening at around 8pm. This pattern reflects both the emissions activity and the mixing status of the atmosphere throughout the day. The predicted "best-fit" $N_{10}$ concentration follows this same pattern. Nucleation continues to play a role during winter but does not dominate to the point that it produces a midday peak in $N_{10}$ concentrations . Non-residential natural gas combustion is predicted to be the largest source of $N_{10}$ during morning and evening peaks. The diurnal profiles of non-residential natural gas emissions are included in supplemental information (Figure S2) along with the regional distribution of those emissions (Figure S3). These diurnal variations of the natural gas combustion emissions were obtained directly from the emissions inventory specified by the California Air Resources Board. Industrial natural gas combustion emissions peak during the daytime with lower values at night. Emissions from electricity generation powered by natural gas peak in the morning and evening. Commercial natural gas combustion emissions may either peak in the morning and evening or they may follow a uniform diurnal profile depending on the specific source and location.

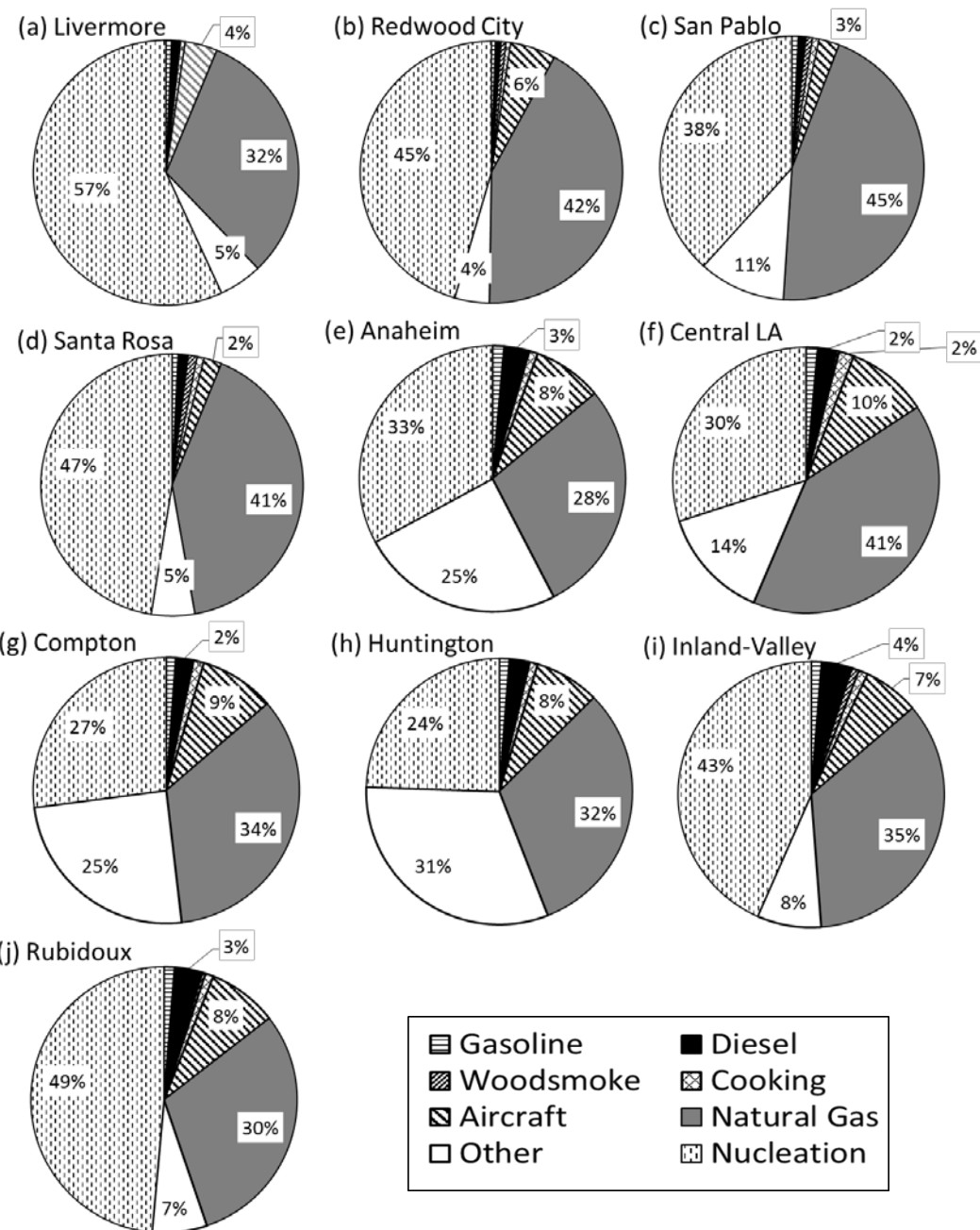

Figure 13: The relative source contributions to $N_{10}$ at Livermore, Redwood City, San Pablo, Santa Rosa, Anaheim, Central LA, Compton, Huntington, Inland-Valley and Rubidoux, respectively.  Averaging time included all days shown in Figures 7-9.  Values not displayed are ≤ 1%.

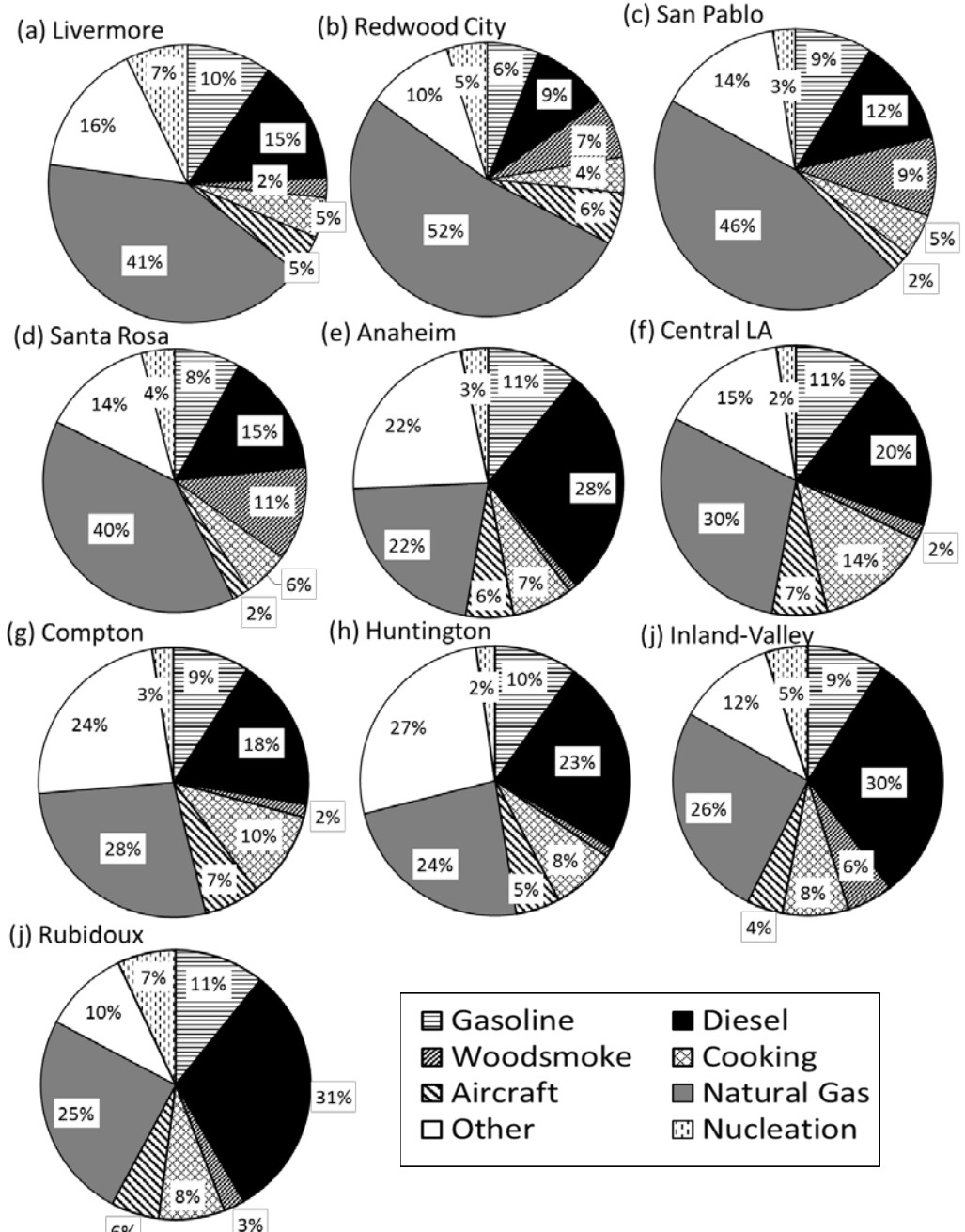

Figure 14: The relative source contributions to PM$_{0.1}$ seasonally averaged at Livermore, Redwood City, San Pablo and Santa Rosa, Anaheim, Central LA, Compton, Huntington, Inland-Valley and Rubidoux, respectively. Averaging time included all days shown in Figures 10-12. Values not displayed are ≤ 1%.

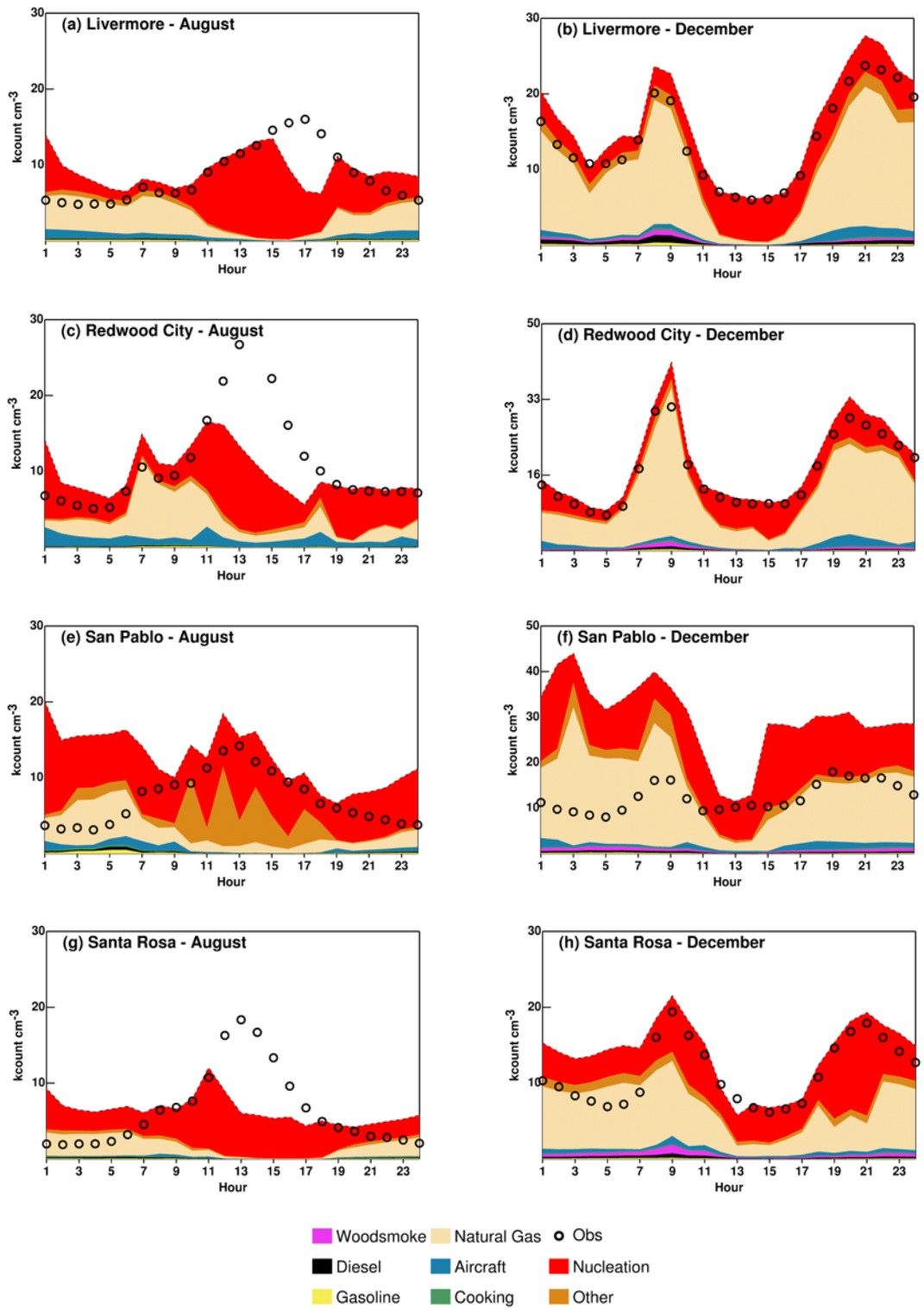

Figure 15: Diurnal variations of measured $N_7$ and predicted "best-fit" $N_{10}$ averaged for August 2012 (left column) and December 2012 (right column) at Livermore, Redwood City, San Pablo and Santa Rosa.

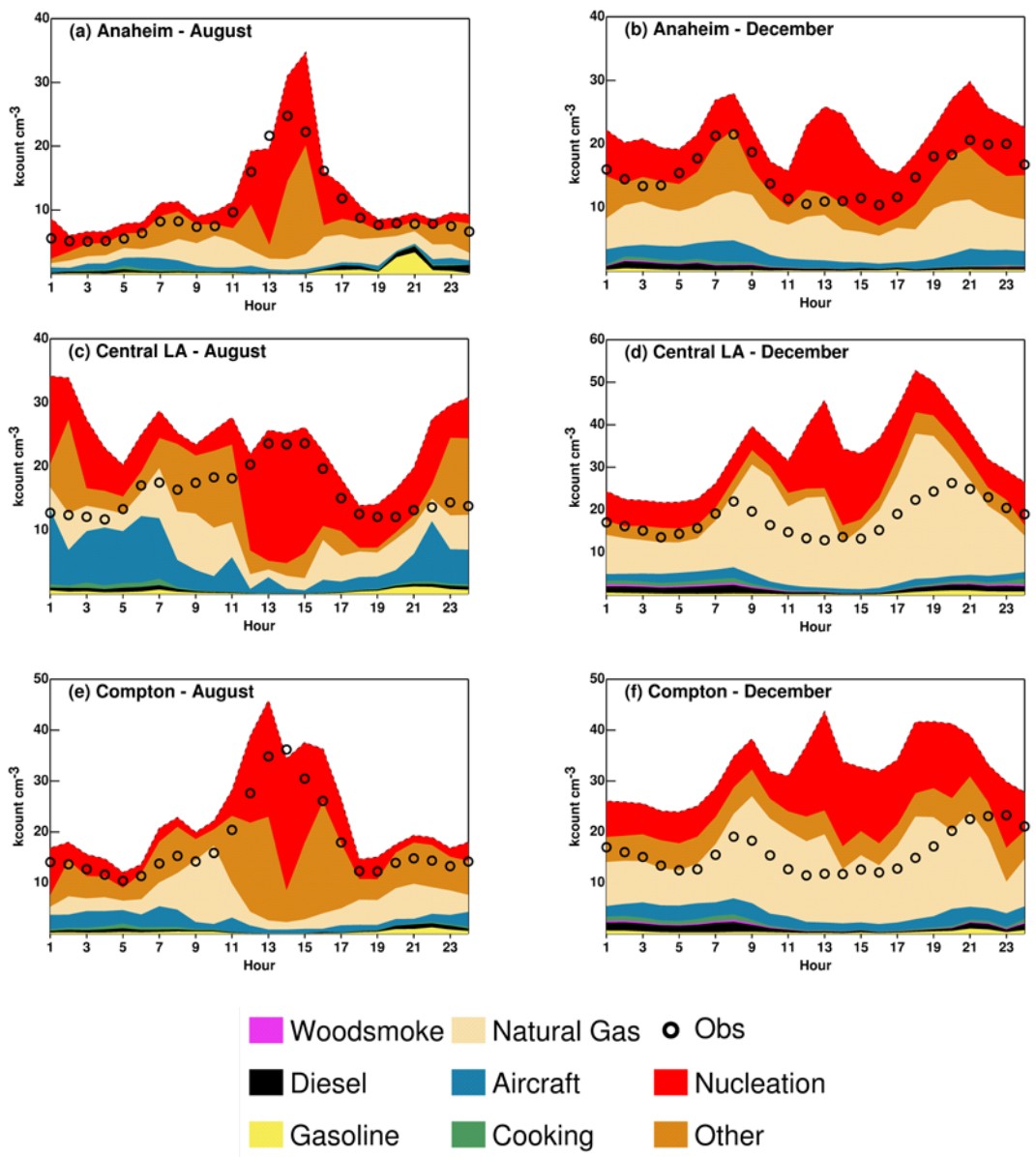

Figure 16: Diurnal variations of measured $N_7$ and predicted "best-fit" $N_{10}$ averaged for August 2012 (left column) and December 2012 (right column) at Anaheim, Central LA, and Compton.


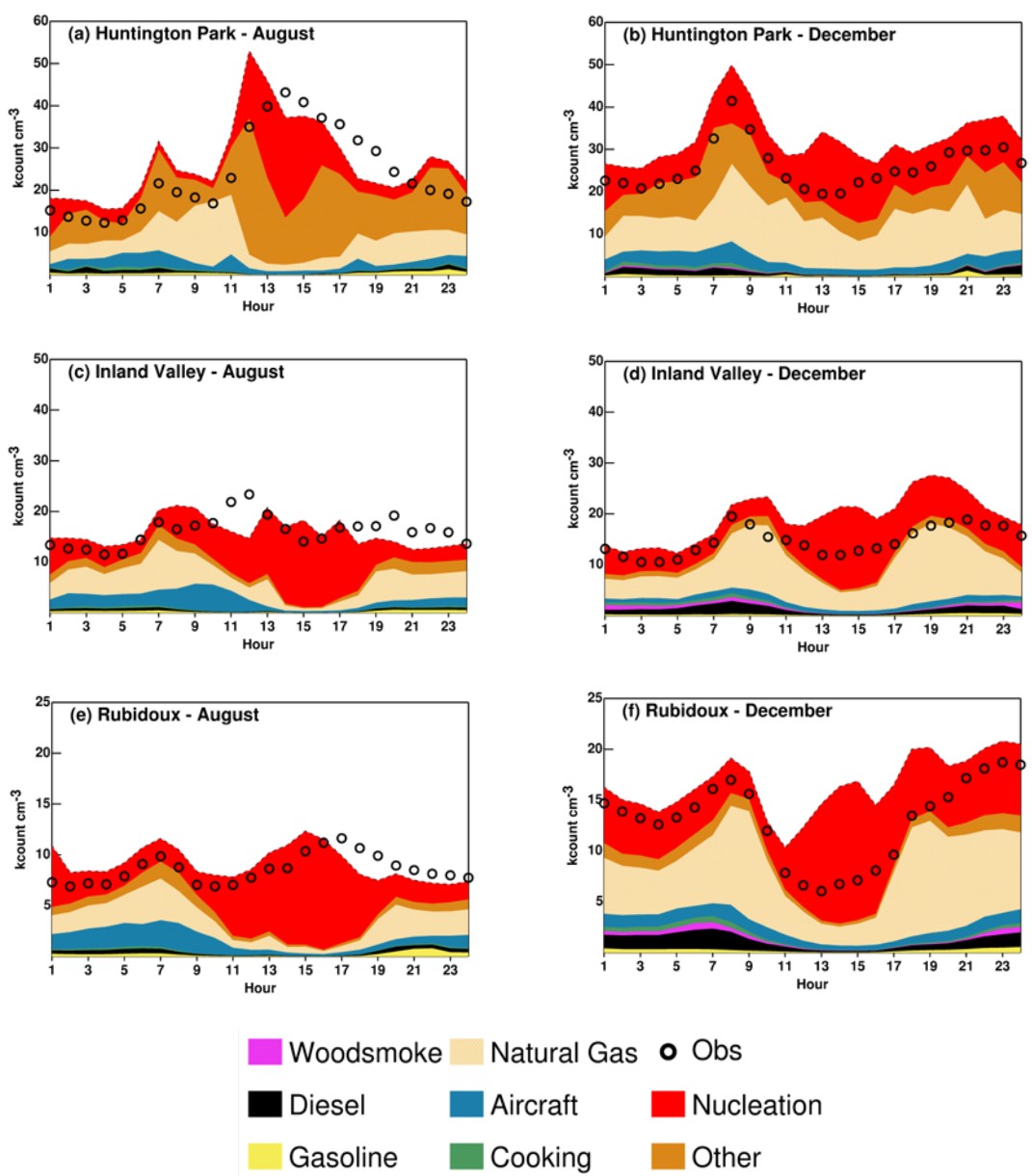

Figure 17: Diurnal variations of measured $N_7$ and predicted "best-fit" $N_{10}$ averaged for August 2012 (left column) and December 2012 (right column) at Anaheim, Central LA, and Compton.


### 3.2.3 Regional $N_{10-1000}$ Source contributions in California

Figure 18 illustrates the predicted number concentration associated with primary emissions (Figures 18a-i) and nucleation (Figure 18j) in southern California averaged

over the months Aug-Dec 2012.  Figure 18g shows that primary aircraft emissions in the

plume downwind of the Los Angeles International Airport (LAX) are predicted to

account for 8 kcounts cm$^{-3}$ and Figure 18j shows that nucleation of aircraft emissions in

the LAX plume are predicted to account for 45 kcounts cm$^{-3}$ yielding a total number

concentration associated with LAX aircraft of approximately 53 kcounts cm$^{-3}$.  Hudda et

al. (2014) found that particle number concentrations increased by a factor of four to eight

downwind of LAX based on measurements in June-July 2013.   Total ground-level

number concentrations in the LAX plume reached 60-70 kcounts cm$^{-3}$.  Given the 4km

spatial resolution of the model calculations used in the current study, the predictions and

measurements of particle number concentration downwind of LAX are consistent with

one another.

It is noteworthy that military airbases in Figure 18g have significantly higher particle

number concentrations due to their use of aviation fuel with higher sulfur content, but

nucleation plumes are not present downwind of these locations (Figure 18j).  Particles

emitted from military aircraft are represented as primary emissions in the current model

calculations.  Future measurements should compare particle number concentrations

downwind of civilian and military airports to fully evaluate the impact of aviation fuel

sulfur content on ambient ultrafine particle concentrations.

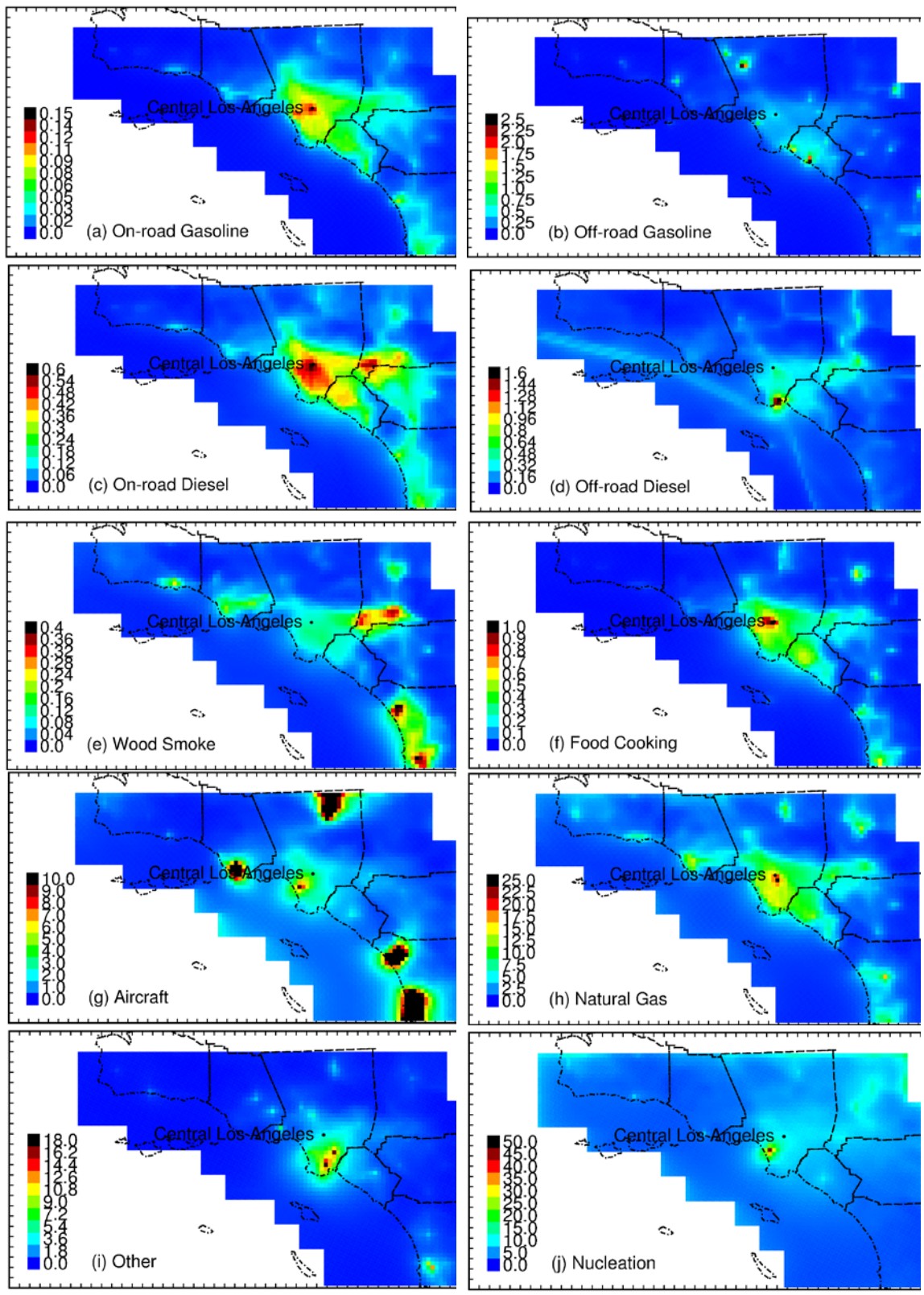

Figure 18. Spatial distribution of particle number from major sources in Southern California (unit: kcount cm$^{-3}$).

Figure 19 illustrates the predicted particle number concentrations associated with primary

       sources and nucleation in northern California.  The relative importance of sources and the

       prediction of nucleation downwind of major sulfur emissions are consistent in northern

       and southern California.  Natural gas combustion is a notable strong source of ultrafine

       particles in both regions due to the widespread use of this fuel in numerous residential,

commercial, and industrial applications.  In many cases, the natural gas combustion

       particles contribute strongly to the "urban background' concentrations over most

       California cities without the formation of individual plumes such as those found

       downwind of LAX.  Future measurements could correlate ambient particle number

       concentrations and natural gas utilization across multiple cities to evaluate whether

natural gas combustion is a significant source of particle number concentration.

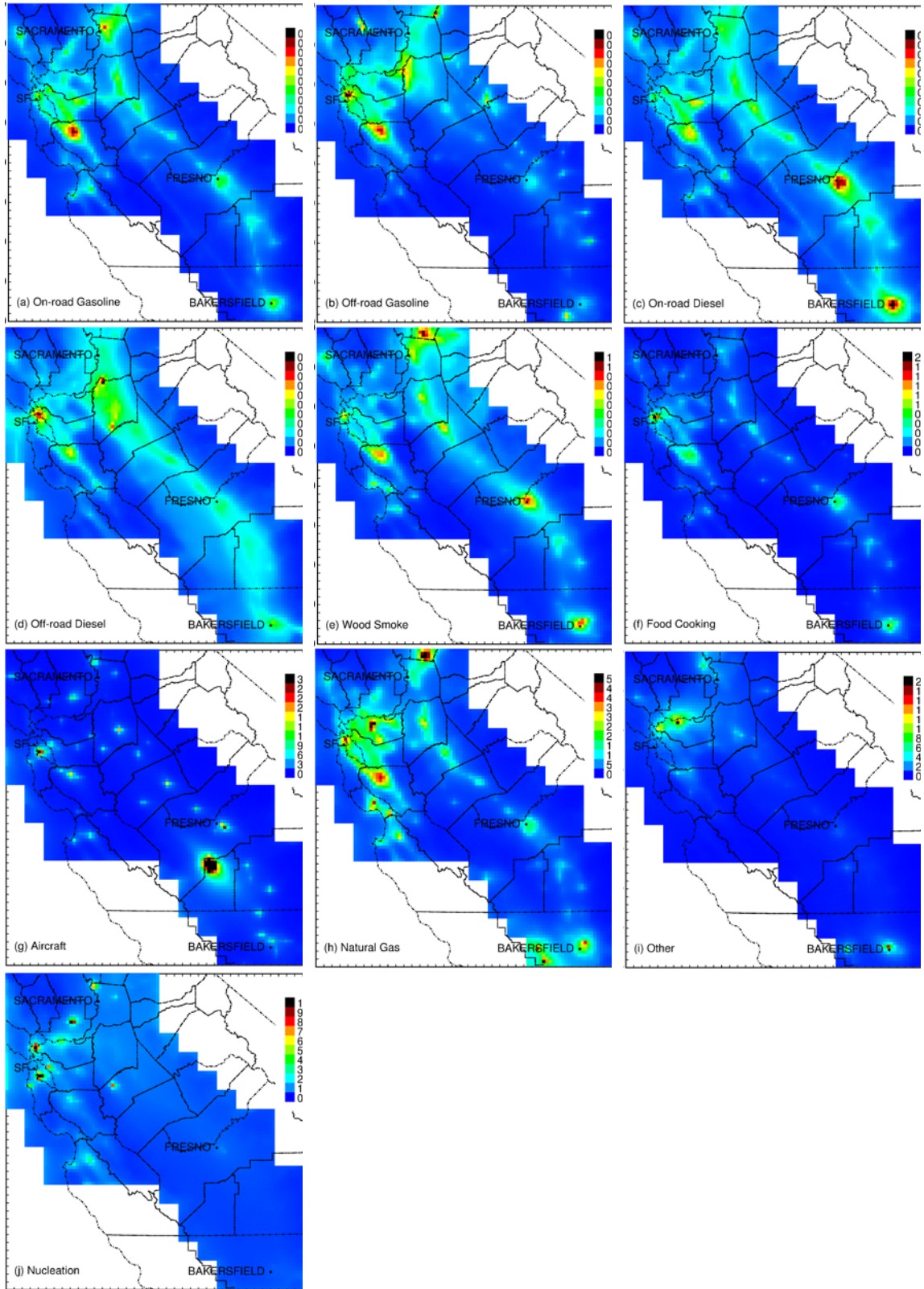

Figure 19. Spatial distribution of particle number from major sources in Northern California (unit: kcount cm$^{-3}$).

The concentrations of nucleated particles in August, October, and December are shown in

Figure 20 (Southern California) and Figure 21 (Northern California) below. Nucleation

events occur in the regions where sulfur emissions are highest (typically airports,

shipping ports and refining facilities). Concentrations of nucleated particles are higher in

October and December than in August because colder temperatures increase nucleation

rates if the precursor $H_2SO_4$ and $NH_3$ concentrations are relatively constant. A significant

fraction of the $H_2SO_4$ in the current simulation is produced by the fast conversion of gas-

phase $SO_3$ emissions to $H_2SO_4$ in the exhaust plume near the emissions source. $SO_3$

conversion does not depend on the presence of oxidants in the atmosphere and so the

higher oxidant concentrations in the summer do not dominate the seasonal nucleation

pattern.

Once $H_2SO_4$ forms in the exhaust plumes, it either condenses onto existing particles

formed from lower volatility compounds in the plume, or it mixes with $NH_3$ in the

background air and nucleates. This process is captured by dilution source sampling

measurements that allow for a few minutes of aging time and so the size-resolved

emissions profiles for many sources already account for the effects of nucleation within

the "near-field" exhaust plume (within a few 10's of meters after emission). $SO_3$

emissions from reciprocating internal combustion engines were therefore set to zero to

avoid double counting the new particle formation downwind of these sources in the

current study. Regular $SO_2$ emissions from these sources were not modified. Emissions

from aircraft jet engines have high exit velocity which promotes rapid mixing with

background air. $SO_3$ emissions were left at their nominal levels (3-4% of total SOx) for

jet engine aircraft in the current study. The consequence of these model treatments is that

predicted concentrations of nucleated particles are highest downwind of LAX, which

agrees with measurements of ambient particle number concentrations (Hudda et al.,

2014).


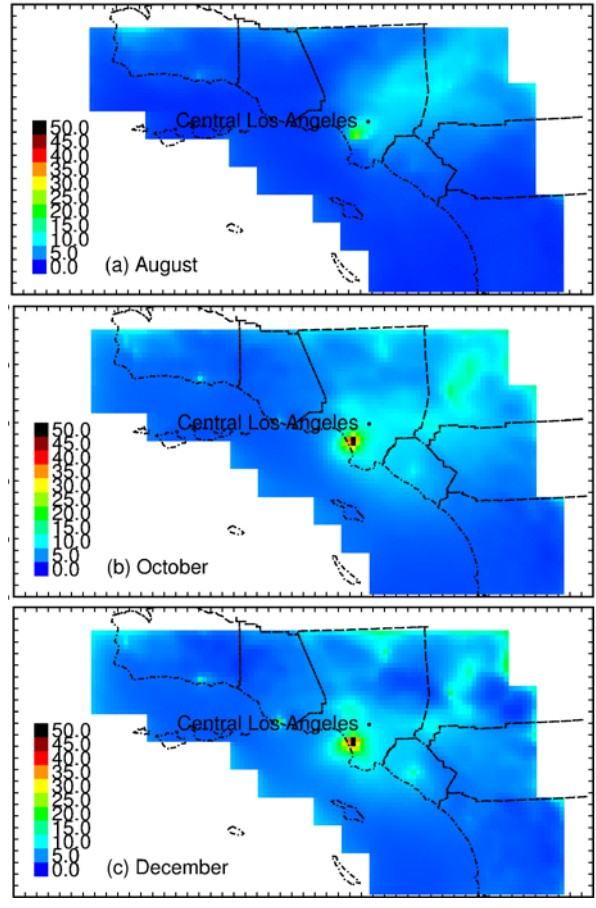

Figure 20: Seasonal variation of nucleated particle concentrations in Southern California. Units are kcount cm$^{-3}$.


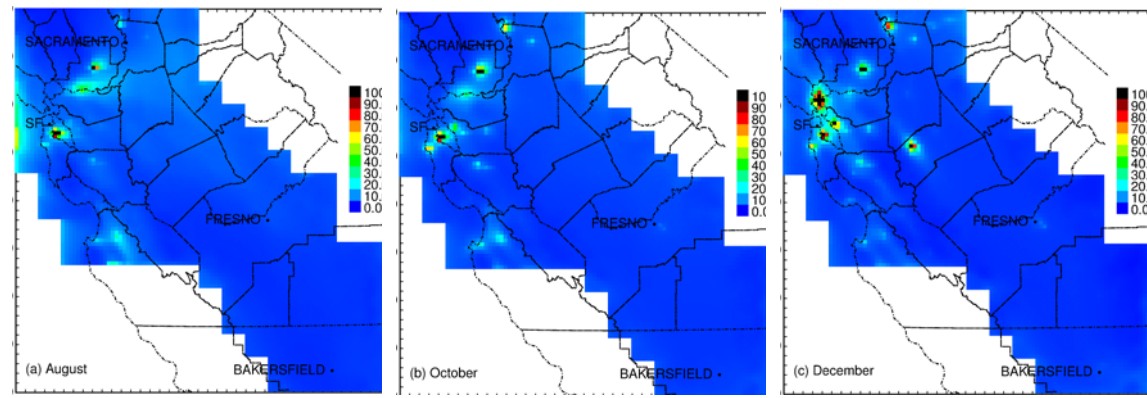

Figure 21: Seasonal variation of nucleated particle concentrations in Northern California. Units are kcount cm$^{-3}$.

**4. Discussion**

Previous researchers have used Positive Matrix Factorization (PMF) to calculate source contributions to $N_7$ (Sowlat et al., 2016;Morawska et al., 2008;Gu et al., 2011;Ogulei et al., 2007;Kasumba et al., 2009;Wang et al., 2013;Yue et al., 2008;Friend et al., 2013). The dominant factors resolved by these studies have been traffic, urban background, secondary aerosol, wood burning and nucleation (Sowlat et al., 2016;Morawska et al.,

2008;Gu et al., 2011;Ogulei et al., 2007;Kasumba et al., 2009;Wang et al., 2013;Yue et al., 2008;Friend et al., 2013). Particles from natural gas combustion were not separately identified by PMF because they do not contain a unique chemical tracer.  It is very likely that natural gas combustion particles are artificially lumped into another source (e.g. traffic) or part of the "urban background" signal identified in previous studies. Natural

gas combustion is used extensively in California for electric power, industrial, commercial and residential use (Table S6), and so it seems plausible that this source contributes to ambient UFP concentrations.

     The current UFP predictions rely on source profile measurements for wood burning, food cooking, mobile sources, and non-residential natural gas combustion (Cooper,

1989;Harley et al., 1992;Hildemann et al., 1991a;Hildemann et al., 1991b;Houck and L. C., 1989;Kleeman et al., 2008a;Kleeman et al., 2000;Robert et al., 2007b;Robert et al., 2007a;Schauer et al., 1999b, a, 2001, 2002a, b;Taback, 1979). All of these size distributions were measured using appropriate instruments and methods by knowledgeable researchers, but some of these past studies were conducted more than a

decade ago. Size distribution information for vehicles, natural gas, etc. have been added to the supplemental information (Figure S4). Changes in fuel composition and emissions control technology in the interim years may have altered the emitted size distributions. New measurements of particle size distributions emitted from natural gas and biomethane combustion were made in parallel with the current project to confirm the source profile

measurements from past studies (Xue et al., 2018a).  The results of these measurements are consistent with previous size distribution results (Li and Hopke, 1993).

     California has tighter air pollution standards than many other regions in the United States due to the severe air quality problems that have historically occurred in the state.

California therefore has a unique combination of fuels and emissions control technology

that may affect the mixture of sources that contribute to atmospheric ultrafine particle

concentrations.  Venecek et al. (2018) recently used the UCD/CIT air quality model with

the 2011 National Emissions inventory to calculate source contributions to $PM_{0.1}$ in 39

major cities across the United States during peak summer photochemical smog episodes

in the year 2010.  The findings from this study show that natural gas combustion is a

major source of ultrafine particles in the regional atmosphere over urban areas across the

United States.  The public health questions associated with ultrafine particles emitted by

natural gas combustion have wide-ranging implications.  Similar levels of ultrafine

particle concentrations will likely occur in other regions across the world that extensively

use natural gas as a fuel source, although other sources of ultrafine particles may also

make strong contributions depending on the total mix of fuels in each region.

Recent theories suggest that primary particulate matter composed of semi-volatile organic

compounds may evaporate after release to the atmosphere, which may reduce ambient

$N_x$.  Measurements conducted in parallel with the current study confirmed that particles

emitted from natural gas combustion in home appliances partially evaporated when

diluted by a factor of 25 in clean air, but particles emitted from reciprocating engines did

not evaporate under the same conditions (Xue et al., 2018a).  Future work should verify

the accuracy of the size and composition distributions for all natural gas combustion

sources given their apparent importance for predicted $N_x$.

Evidence from both toxicology and epidemiology will be required to assess the effect of

UFPs on public health.  It is essential to identify and quantify UFP sources based on both

mass ($PM_{0.1}$) and $N_x$ during this process (Friend et al., 2013). An accurate comparison of

both $PM_{0.1}$ and $N_x$ exposure could lay the groundwork for specific assessment of health

effects of UFPs and potentially more efficient control strategies for PM emission from

major sources (Yue et al., 2008). Ideally, spatial exposure patterns for $N_x$, $PM_{0.1}$, and

$PM_{2.5}$ will be sufficiently unique to separate their individual effects in epidemiological

studies.  Regression statistics for different metrics were calculated by using all grid cells

in the model domain of the current study. The correlations between the various particle

metrics were: $R^2(PM_{2.5}$ vs. $N_{10})=0.35$, $R^2(PM_{2.5}$ vs. $PM_{0.1})=0.63$, $R^2(PM_{0.1}$ vs. $N_{10})=0.75$.

It seems likely that future epidemiological studies will be able to differentiate between

the effects of $PM_{2.5}$ and $N_x$ based on the low $R^2$ value.  The potential for comparisons between $PM_{2.5}$ and $PM_{0.1}$ is less clear cut, but previous work helps understand what may be possible.  Ostro et al. (2015) compared the associations between IHD mortality and $PM_{2.5}$ vs. $PM_{0.1}$ in the California Teachers Study (CTS) cohort.  Associations between IHD mortality and the sum of $PM_{2.5}$ mass (p-value=0.001) were stronger than

associations between IHD mortality and the sum of $PM_{0.1}$ mass (p-value=0.01) but individual components of mass (EC, OC, Cu, etc) all had stronger associations with IHD mortality in the $PM_{0.1}$ size fraction than the $PM_{2.5}$ size fraction.

The current study focuses on outdoor exposure to UFPs that may be useful in future epidemiological studies.  Indoor or in-vehicle exposure to UFPs can also be significant

(Wallace and Ott, 2011;Rim et al., 2010;Bhangar et al., 2011;Weichenthal et al., 2015;Fruin et al., 2008) but characterizing these micro-environments is beyond the scope of the current manuscript.

**5. Conclusions**

The UCD/CIT regional chemical transport model has been updated with a nucleation

algorithm and combined with the existing size-resolved source profiles of particlualte matter emissions to predict regional source contributions to airborne particle number concentration ($N_{10}$) and airborne particulate ultrafine mass ($PM_{0.1}$).  Predicted 24-hour average $N_{10}$ follows the same trend as measured $N_7$ at ten sites across California in summer (Aug) and winter (Dec). Predicted diurnal variation of $N_{10}$ follows the same

trend as measured concentrations at the majority of the evaluation sites in August and December, but the results suggest that further refinement is needed for both primary emissions and nucleation algorithms.   Predicted $PM_{0.1}$ source contributions follow the same trends as $PM_{0.1}$ source contributions measured in a molecular marker study at four sites across California in summer (August) and winter (December) months.  Natural gas

combustion is the largest primary source of regional $N_{10}$ at all locations outside of the immediate vicinity of other major combustion sources.  Nucleation contributed strongly to particle number during both the summer and winter months.  Traffic sources

contributed to $N_{10}$ but did not dominate over regions more than 300 m away from freeways. Combustion sources such as wood burning, food cooking, and mobile sources made stronger contributions to $PM_{0.1}$ at heavily urbanized locations.  Wood burning for home heating had strong seasonal patterns with peak concentrations in winter while other sources contributed more consistently throughout the seasons.  Nucleation made a negligible contribution to $PM_{0.1}$ over the urban areas at the focus of the current study.

The current study identifies natural gas combustion as an important source of ultrafine particle number and mass concentrations in urban regions throughout California.  The health implications of these natural gas combustion particles should be investigated in future epidemiology studies.

Data Availability: All of the $PM_{0.1}$ and $N_x$ outdoor exposure fields produced in the current study are available free of charge at http://faculty.engineering.ucdavis.edu/kleeman/ which provides a link to the most recent version of the dataset (currently http://webwolf.engr.ucdavis.edu/data/soa_v2/monthly_avg2).  Model source code and model input files are available to collaborators via direct email to the corresponding author at mjkleeman@ucdavis.edu.

**Acknowledgements:** This research was supported by the Bay Area Air Quality Management District under project #2013.218, the Coordinating Research Council under project #A-96, and the California Air Resources Board under project #14-314.  None of the project sponsors nor any person acting on their behalf: (1) makes any warranty, express or implied, with respect to the use of any information, apparatus, method, or process disclosed in this report, or (2) assumes any liabilities with respect to use, or damages resulting from the use or inability to use, any information, apparatus, method, or process disclosed in this report.

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
