# Peer review of "Regional Sources of Airborne Ultrafine Particle Number and Mass Concentrations in California"

_Atmospheric Chemistry and Physics, 2018_

## Referee Comment (RC1) · Anonymous Referee #3 · 8 Nov 2018

The present paper attempts to quantify the sources of ultrafine particle number and mass in California using a chemical transport model. Determination of the sources of particle number is a challenging problem that has received a lot of attention for global scales, but relatively little for urban and regional scales. The work arrives at a rather controversial conclusion that contradicts most existing studies (including Hu et al. (2017) by the same team): natural gas combustion is the largest source of particle number and a significant source of ultrafine mass at most locations and periods at least in California. The paper is rather uneven: it does a good job dealing with the sources of ultrafine mass, but at the same time its treatment of the sources of particle number is problematic and potentially wrong. Its major weaknesses do not allow me to recommend its publication in its current state. However, the paper does offer important

new insights and should be publishable if its weak parts are corrected or even deleted. The various issues that need to be addressed are described below.

(1) Major problems

(1.1) Sources of particle number. The authors calculate the particle number contributed by each source based on the corresponding mass (equation 1 in the paper). This is wrong for two reasons. First a significant fraction of the particle mass is secondary (sulfates, nitrates, secondary organic aerosol). When the secondary mass increases, the contribution of the corresponding source to particle number does not. Second, co-agulation involves particles from different sources. It is not clear to which source the authors assign the particle resulting from the coagulation of two particles from different sources. Both of these problems are quite important for ultrafine particle number concentrations. The errors of this oversimplified approach should be estimated (at least for one period) with careful zero-out analysis (e.g., removing only the ultrafines and not the larger particles to avoid changes in the condensation and coagulation sinks). If the error is significant the corresponding part of the work should be redone or should be replaced with just a description of the contributions to emissions for different size ranges.

(1.2) Importance of non-residential natural gas combustion as a source of ultrafine particles. This is clearly the most important, but also the most controversial finding of the study. The evidence provided to support this potentially very important result is rather weak and the authors miss a lot of opportunities to strengthen their argument.

The first is the use of size distributions. The predicted size distributions from this source apparently peak in the 10-20 nm size range. There are a lot of available size distribution measurements in the area that can be directly compared with the model predictions. My understanding howeveris that the measured number size distributions (not imme-diately next to freeways) peak at the 35-40 nm range (see for example Sowlat et al., 2016). Some of these size distribution measurements are available for the periods that

have been simulated so a comparison of size distributions (including sources) could be performed without much effort.

The second is the use of the spatial distribution of particle number. The predicted concentration maps are not shown, but one would expect much higher concentrations near the corresponding major source areas. Traffic should have quite a different spatial pattern. There have been also a lot of particle number distribution measurements in California during the last decade. An effort to test if the predicted patterns match the observed ones would help.

The third is the average diurnal variation. However, this study assumes that the non-residential natural gas emissions have a similar temporal pattern as traffic (Figure S2). So the observed rush-hour peak in particle number that all previous studies assign to traffic, here is explained by natural gas combustion. However, more careful spatio-temporal analysis could help strengthen (or weaken) the conclusion. For example, the predicted morning number peak in Roubidoux in summer does not exist in the measurements. The situation is even worse in midday during the winter suggesting that emissions from this source are clearly overestimated in this area. Is this helpful? Is this area dominated by these emissions or is the sampling site an exception? On the other hand, the model performs well in other areas so one could make the opposite argument site by site. However, without using all the information about predicted patterns in space and time it is difficult to reach a conclusion.

(1.3) Modeling of growth of ultrafine particles. The approach used to simulate condensation/evaporation of sulfuric acid, ammonium, nitric acid, secondary organics on the ultrafine particles in this study is not explained in any detail. There is a rather confusing statement in lines 129-137 that "dynamic condensation/evaporation is not considered". Does this mean that the particles are assumed to be in equilibrium? If yes, how does the model deal with the effect of surface tension on the equilibrium vapor pressure especially in the 10-20 nm range? Do these particles evaporate because their equilibrium vapor pressure is higher than that of the bigger particles? This is a crucial process for

the number concentration of the smaller particles and it is not clear that it is simulated properly.

(2) Other significant issues

(2.1) Definition of particle number concentration. The use of the term particle number concentration throughout this paper is often confusing and sometimes misleading. It is important to always define the lower threshold of the size range of the corresponding concentration. The total particle concentration can be easily a factor of 2 or 3 higher than the concentration of particles with diameter higher than 10 nm (N10).

(2.2) Growth of freshly nucleated particles to 10 nm. The authors state that they parameterize the growth process following the work of Kerminen and Kulmala (2002). However, this parameterization requires the growth rate (GR) of the particles. The calculation of this rate is non-trivial in a model with coarse aerosol size resolution such as the current one. Errors in the GR can lead to significant errors in the estimation of the contribution of nucleation as a source to particle number. The authors should evaluate the error of this parameterization for their aerosol model.

(2.3) There is little information provided about the frequency and spatial extent of nucleation in the simulations in the various seasons. This information is needed to understand the simulation results.

(2.4) Emissions from natural gas combustion. A map of the estimated N10 and PM0.1 emissions from this major source is needed (see also comment 1.2). Also the average diurnal profile of the emissions for the domain and the average size distribution should be shown.

(2.5) Temporal scale of evaluation. The authors present metrics of the model performance but they do not clarify if these are for hourly, daily, monthly, simulation averages or something else. The text and the corresponding tables do not include this information. Given the availability hourly measurements evaluation at this timescale should be

also performed (if it has not been performed yet). The evaluation at a daily scale is also useful.

(2.6) The measurements of particle number refer to N6 and N7 while the predictions to N10. The authors suggest that the average error in the corresponding comparisons should be less than 10

(2.7) The use of qualitative terms (general agreement, agree reasonably well, good agreement) is not helpful and should be avoided.

(2.8) If my understanding of the paper is correct, the current model does not use the dynamic organic aerosol scheme used by Hu et al. (2017). If this is the case, the results regarding the contribution of SOA to PM0.1 in this work should be discussed and should be compared to that version of the model. If it is the same it should be clearly stated.

(2.9) Contribution of traffic particles. Ronkko et al. (PNAS, 114, 7549-7554, 2017) argued that traffic is an even more important source of particle number, because there are a lot of sub-10 nm particles emitted. Given that the current study does not include primary traffic particles smaller than 10 nm (which of course can grow to larger sizes), can it seriously underestimate the contribution of this source in urban environments?

(3) Minor issues

(3.1) The abstract says that simulations have been performed for 2012, 2015 and 2016. However, the presented results are only for 2012. This is important because there are available size distribution measurements for 2015-16 in the modeling domain that can be used for the evaluation of the model predictions (see comment 1.2).

(3.2) The predicted correlations between PM2.5 and particle number concentrations can be compared with the corresponding measured correlations as an indirect way to evaluate the model performance.

(3.3) Lines 66-67 "when nucleation algorithms were not standardized". This statement

is confusing.

(3.4) Are the sulfate and nitrate concentrations shown in Table S4 for PM2.5 or for another size range?

(3.5) Table 1 should probably also include the predicted and measured average number concentrations.

(3.6) The terms "measured" and "predicted" should be used everywhere in Section 3.2.1 and other parts of the paper in which predictions are compared to measurements.

(3.7) The number of samples and their duration corresponding to the results of Figs. 2-3 should be stated in the caption.

(3.8) Line 296. Figures 4-6 and 7-9 do not show the seasonal variation of the corresponding variables. They show data (are these daily averages or something else) for different days in different seasons. These figures could be improved if they were split in four parts for the different periods simulated. The discussion could also be improved if the actual seasonal averages were shown (may be in the SI) and discussed.

(3.9) Figure S2. What is A, B, and C? What is the average pattern in the domain?

---

## Referee Comment (RC2) · Anonymous Referee #2 · 23 Dec 2018

**Summary:**

This work demonstrates the source appointment of ultrafine particle number and mass concentrations in California using the UCD/CIT chemical transport model. The manuscript fits well to the scope of ACP. However, I am worried about the method used for retrieving PN source and corresponding conclusions. This paper is worth to be published, but not in its current form. Thus I recommend it to be resubmitted after the following major comments listed below have been adequately addressed.

**Comments:**

1. Maybe MFE and MFB are very useful to present the model performance. But I would suggest the authors also provide the correlation between the predicted and measured results, which is more straightforward.

2. Page 8, line 230: I am confused why the author raise the value of 8% ($N_{7-10}/N_{7-1000}$) here? Did you use it to correct model results? If so, then this value is measured at Fresno supersite, which is located near roadways with moderate traffic. So could it be used for all the cases? Also, the particle number concentration has a significant diurnal variation, especially during the nucleation event days. But the authors only compare the daily average, this might be problematic. Concerning the particle number simulation, the number size distribution is also very important. Do they have any number size distribution measurements on the sites? I think it might be worth to compare.

3. Page 10,equation 1: The method used to convert mass contribution to number contribution is questionable. First, which mass (mass size distribution or total mass) do you use in eq. (1)? It is not clear how you define the Dp. Second, the nucleation is a major source of particle number, but it won't contribute a lot to the mass concentration, so if you use the mass size distribution in eq. (1), then it is better to check the number size distribution of nucleation source to evaluate the method. Also, condensation is an important process for the growth of nucleation mode particles. So the change of density can not be ignore.

4. Page 12, line 268: which method was used to measure $PM_{0.1}$ in Xue's paper? In figures 2-3, the authors only compare the data in 2015 and 2016. I guess there was no measurement in 2012. But in Figures 7-9, you only show the time series of $PM_{0.1}$ in 2012? This selective comparison is also shown for particle number concentration (figures 4-6). I would suggest the authors should also show the time series data in 2015 and 2016, which contain both measurement and modelling results. Additionally, what is the time resolution in Figures 4-9? It seems the x-axis in Figures 4 and 7 is not regular.

5. In Figures 2-3, there is no nucleation source, if the authors use eq.1 to convert mass contribution to number contribution, then wow did the authors define "nucleation" source?

6. Nucleation is a major source of particle number concentration. I would suggest the authors also show the modelling results only for nucleation days. If you put it in the average data (figures 12-14), then more information might be covered. And why you only show the average data from August and December.

---

## Referee Comment (RC3) · Anonymous Referee #1 · 30 Dec 2018

Yu et al., use the UCD/CIT model to simulate ultrafine particulate matter in California, focusing on the Los Angeles and San Francisco areas. To do so, they have developed an inventory of relevant emissions and added a nucleation model to the code. They find acceptable model performance. A particular finding is that non-residential gas combustion is a dominant contributor.

The paper has some interesting aspects to it, particularly the assignment of sources to their impacts on particle number. This may also be its weakness as there is little means to assess the validity of some of the resulting conclusions that might be drawn and the results are striking and don't really line between the model simulations and the observations. Further, they don't bring in recent findings.

Their main result is that non-residential natural gas (NRNG) combustion is the major

contributor to particle number often contributing over half. Looking at Fig. 10, NRNG contributes about 60-70% of the total at almost all the cities (slightly more at Rubidoux, somewhat less at Livermore). This is remarkably consistent given what has been found about the contribution of mobile sources and aircraft emissions to UFPs in other studies (e.g., U Wash, USC studies). They don't include aircraft in these plots: this is a huge shortcoming, and on this alone, the manuscript requires much more work before being considered for publication. A major weakness here is also that the emissions from NRNG, vs. residential NG, is from a recently published manuscript. However, in my reading of that manuscript, they do not include the conditions referred to in this manuscript (a dilution factor of 25), and they seemed to focus on biogas. Maybe the use of the word "same" is of issue here as well. It should be noted that the observations also do not support that the main source is NRNG (and their model results suggest this as well), as particle number increases at night in December, starting about rush hour and going until about 8 pm. This very much looks like mobile source emissions, but certainly not an industrially-related source that would likely decrease after 17:00. During the summer, there appears to be more of a mid-day, photochemically-generated peak. Overall, the observations tend to suggest something very different than the model.

They make the statement that "traffic sources contributed to PNC but did not dominate over regions more than 300 m away from freeways." This is a rather strange statement given that their model resolution is 4 km. They have no way of supporting this statement. Their making this statement is worrisome.

They also state in the Abstract that the performance meets the threshold normally required for regulatory modeling. I am not aware that such a threshold has been set. I don't believe the Boylan and Russel paper is accepted by any agency. Further, they need to be much more informative as to how they actually calculated the performance statistics given that the number concentrations are available at a finer time resolution than the species concentrations often used in performance determinations. Maybe they should also look at the AQMEII studies. The current table of performance (Table

1) is insufficient.

Their reference to Shet et al., referring to Taylor's hypothesis, is not relevant here. Taylor was looking at turbulence correlations, and the relationship between temporal fluctuations and spatial fluctuations. Here, one has to assume that emissions and chemistry play a huge part, particularly since the observations are averaged over scales much larger than the Taylor scale. It was not even apparent why they cited the paper.

Looking at Fig. 12, there are a number of locations where there appears to be a mismatch between 23:00 and 0:00.

Boundary conditions can be very important in regions close to the coast. A diagram of the modeling domain should be provided along with the boundary conditions. A test of the impact of boundary conditions on the results should be provided.

Their modeling domain height is only 5 km. This is lower than most any other model used, from what I recall. Citing some of their studies without really doing a comparison as to the impact of having a higher domain is not sufficient.

What is meant by "Model source code and model input files are available to collaborators via direct email... It should be made available to anyone looking to check their results. A more general statement of availability should be provided. All files and data needed to recreate the results should be available.

Fig. 3. Two issues here. First, the caption suggests that both CMB and UCD results are shown. Are CMB results labeled as "Obs.". This would be a wrong interpretation. Further, how are the uncertainties determined? Second, they should also show secondary fractions.

Figs. 5-6, a correlation plat would be useful. The obs seem to be rather less variable. Fig. 10-11. These results bring up a question: Were the same size distribution on the emissions used everywhere on a source-by-source basis.

Line 514: It should be "under".

Summary: At present, there a serious issues with the paper, including not including aircraft impacts, that the result that NRNG is the dominant contributor does not appear to explain the observations, some statements that are off-base (scale of impact of freeways, performance metric for regulatory acceptance, Taylor's hypothesis) and the need to better describe how performance was evaluated. A major rewrite, alone, may not be able to address all of the concerns.

---

## Author Comment (AC1) · 10 Jul 2019

Reviewer 1 Comment 1: Sources of particle number. The authors calculate the particle number contributed by each source based on the corresponding mass (equation 1 in the paper). This is wrong for two reasons. First a significant fraction of the particle mass is secondary (sulfates, nitrates, secondary organic aerosol).  When the secondary mass increases, the contribution of the corresponding source to particle number does not. Second, co- agulation involves particles from different sources. It is not clear to which source the authors assign the particle resulting from the coagulation of two particles from different sources. Both of these problems are quite important for ultrafine particle number con- centrations.  The errors of this oversimplified approach should be estimated (at least    for one period) with careful zero-out analysis (e.g., removing only the ultrafines and not the larger particles to avoid changes in the condensation and coagulation sinks). If the error is significant the corresponding part of the work should be redone or should be replaced with just a description of the contributions to emissions for different size   ranges.

Response: We apologize that the methods to calculate particle number were not explained clearly in the first version of the paper.  The model framework uses a moving sectional approach to conserve particle number and mass while letting particle radius increase due to condensation (Kleeman, Cass, and Eldering 1997).   The method to calculate source contributions to number concentration is performed for each moving section individually.  Number is explicitly conserved and correctly apportioned to sources in this algorithm.

Each particle source type / moving size bin includes an artificial tracer equal to 1% of the primary particle mass.   The mass of this tracer is related to the number of particles by the equation

Tracer_source_i * 100 = N_source_i * 3.14159/6 * Dp_bin * density_source_i

This equation can be easily rearranged to solve for N_source_i as a function of Tracer_source_i in each size bin.  Again, since the model uses a moving sectional approach, number and tracer mass are exactly conserved.  Condensation/evaporation changes the particle diameter as semi-volatile components move on and off the particle but this does not change Tracer_source_i or N_source_i.  The moving sectional approach greatly simplifies the source apportionment of particle number compared to other models that use fixed particle size bins with condensation / evaporation transferring material between bins.

Coagulation is fastest between very small particles and relatively large particles in the atmosphere.  The net effect of coagulation is to remove ultrafine particles from the atmosphere as they collide and join the particles larger than 100 nm.  This loss mechanism is accurately simulated in the model calculations.  The rate of "self-coagulation" between two ultrafine particles that produces a particle still in the ultrafine particle size range is negligible at atmospherically relevant concentrations.  Table 2 compares the timescale for 0.01 µm particles coagulating with other 0.01 µm particles and coagulating with 0.1 µm particles based on size distributions measured in a typical suburban environment in a California city (see Figure 1).  The coagulation of timescale between two 0.01 µm particles is 209 hrs (8.7 days) while the coagulation timescale between 0.01 µm particles and 0.1 µm particles is 4.4 hours.  Therefore, self-coagulation between ultrafine particles smaller than 60 nm is much less significant than coagulation between ultrafine particles and larger particles (acting as a loss mechanism for ultrafine particles in the atmosphere).

Table 2 Time scale for coagulation between 0.01 µm particles with 0.01 µm particles and 0.1 µm particles in a typical suburban environment in California.  See Fig 1 for size distribution used for calculations.

| Particle size | 0.01 | 0.1 |
|---|---|---|
| Coagulation Coefficient(cm³/s) | 1.90E-09 | 2.50E-08 |
| PM Number concentration (#/cm³) | 1.40E+03 | 5.00E+03 |
| timescale (hours) | 208.9 | 4.4 |

[Figure]

Figure 1 Particle size distribution measured in a typical suburban environment in a California city

Source apportionment calculations treat coagulation events between very small particles and very large particles in a manner analogous to condensation.  When two particles coagulate, the mass of the smaller particle is added to the mass of the larger particle.  The number concentration of the smaller particle is discarded while the number concentration of the larger particle stays constant. This slightly reduces the accuracy of source apportionment calculations for particle number in the larger size bins because the Tracer_source mass in the larger size bin is no longer proportional to the number concentration from that source.  This issue is relatively minor since size bins larger than 1µm that act as the dominant sink during particle coagulation events typically account for less than 5% of the total number concentration.

Perturbation studies were conducted as requested by the reviewer by setting the UFP emissions for on-road gasoline vehicles to zero during the month August 2012.  Emissions of gases and emissions of larger particles from on-road vehicles were not changed.  The difference between this perturbation simulation vs. the basecase simulation was calculated to estimate the number concentration of particles associated with on-road gasoline vehicles.  This "zero-out" concentration is then compared to the standard model source-apportionment calculations in Figure 2 below.  The two methods for number source apportionment yield very similar spatial patterns and very similar maximum concentrations of approximately 0.5 kcounts/cm³.  The tracer source apportionment method accounts for all particle sizes

which produces slightly higher concentrations than the zero-out method that only considered particles smaller than 100 nm. This test confirms that the online source apportionment methods for number in the current study work correctly.

[Figure]

Figure 2 Particle number concentrations associated with on-road gasoline vehicles calculated using the zero-out method and the artificial tracer method in August 2012.

Reviewer 1 Comment 2: Importance of non-residential natural gas combustion as a source of ultrafine particles. This is clearly the most important, but also the most controversial finding of the study. The evidence provided to support this potentially very important result is rather weak and the authors miss a lot of opportunities to strengthen their argument.

The first is the use of size distributions. The predicted size distributions from this source apparently peak in the 10-20 nm size range. There are a lot of available size distribution measurements in the area that can be directly compared with the model predictions. My understanding however is that the measured number size distributions (not immediately next to freeways) peak at the 35-40 nm range (see for example Sowlat et al., 2016). Some of these size distribution measurements are available for the periods that have been simulated so a comparison of size distributions (including sources) could be performed without much effort.

The second is the use of the spatial distribution of particle number. The predicted concentration maps are not shown, but one would expect much higher concentrations near the corresponding major source areas. Traffic should have quite a different spatial pattern. There have been also a lot of particle number distribution measurements in California during the last decade. An effort to test if the predicted patterns match the observed ones would help.

The third is the average diurnal variation. However, this study assumes that the non- residential natural gas emissions have a similar temporal pattern as traffic (Figure S2). So the observed rush-hour peak in particle number that all previous studies assign       to traffic, here is explained by natural gas combustion. However, more careful spatio- temporal analysis could help strengthen (or weaken) the conclusion. For example, the predicted morning number peak in Rubidoux in summer does not exist in the measurements. The situation is even worse in midday during the winter suggesting that emissions from this source are clearly overestimated in this area. Is this helpful? Is this area dominated by these emissions or is the sampling site an exception? On the other hand, the model performs well in other areas so one could make the opposite argument site by site. However, without using all the information about predicted patterns in space and time it is difficult to reach a conclusion.

Response: Xue et al. showed that the primary size distribution for natural gas combustion peaks at approximately 20 nm but the size mode grows to approximately 60 nm after 3 hrs of aging in a smog chamber with a representative urban atmosphere consistent of realistic concentrations of VOCs and NOx under realistic UV intensity (Xue et al. 2018). Similar growth occurs in model calculations meaning that the natural gas particles do not stay static at 20 nm in the atmosphere. The measurements of larger particles in the atmosphere therefore do not definitively identify sources. Expert opinion is still required to interpret the size distributions and assign them to sources. The results of the current study should help refine those expert opinions in the future.

Many of the spatial patterns measured for airborne particle number concentrations have focused on the gradients around roads (see for example (Zhang et al., 2005;Zhang et al., 2004;Zhu et al., 2002a;Zhu et al., 2002b)). Likewise, the study performed by Solwat et al. (2016) referenced by the reviewer was carried out within 150m of a major freeway and so the reported particle size distributions are dominated by traffic sources. These gradients are impossible to resolve using a regional model with 4km resolution. A limited set of additional simulations were conducted using the WRF/Chem model configured with Large Eddy Simulation (LES) around Oakland California so that spatial scales down to 250m could be examined. Maps of the predicted ultrafine particle mass concentrations for gasoline, diesel, food cooking, wood combustion, and natural gas combustion particles are shown in Figure 3 below. At 250m resolution, ultrafine particles from diesel engines peak on major transportation corridors while ultrafine particles from gasoline vehicles are more diffuse reflecting their increased activity on adjacent surface streets. Ultrafine particles from natural gas combustion are even more diffuse reflecting contributions from area sources across the region. As the spatial resolution decreases to 1km and then 4km, the fine details around roadways are artificially diluted in the larger grid cells. This process shifts the dominant source of ultrafine particles over roadways from diesel engines at 250m resolution to natural gas combustion at 4km resolution.

The ultrafine particle model simulations summarized in Figure 3 are consistent with measurements of particle number in the proximity of roadways which show that the traffic contribution to particle number concentration decays to background levels within 300 m (Zhu, Hinds, Kim, Shen, et al. 2002; Zhu, Hinds, Kim, and Sioutas 2002). The measurements made by Zhu et al. indicate that the traffic contribution to regional number concentration cannot be distinguished from other sources on a regional scale using 4km grid cells which is the focus of this study.

Repeating all of the simulations at 250m resolution is beyond the scope of the current study. We emphasize the regional scope of the simulations in the main text of the revised manuscript to inform the readers about the appropriate interpretation of the current results, and we have also added "regional" to the title of the manuscript.

[Figure]

Figure 3: PM$_{0.1}$ mass concentration associated with on-road diesel, on-road gasoline, and natural gas combustion at 250m, 1km, and 4km resolution over Oakland, California.

Hudda et al. (2014) found that particle number concentrations increased by a factor of four to eight downwind of the Los Angeles International Airport (LAX) based on measurements in June-July 2013. Total ground-level number concentrations in the LAX plume reached 60-70 *$10^3$ counts/$cm^3$. Figure 4 illustrates the predicted number concentration associated with primary emissions (Figures 4a-i) and nucleation (Figure j) averaged over the months Aug-Dec 2012. Figure 4g shows that primary aircraft emissions in the LAX plume are predicted to account for 8 * $10^3$ counts/$cm^3$ and Figure 4j shows that nucleation of aircraft emissions in the LAX plume are predicted to account for 45 * $10^3$ counts/$cm^3$ yielding a total number concentration associated with LAX aircraft of approximately 53 * $10^3$ counts/$cm^3$. Given the 4km spatial resolution of the model calculations, these findings are in good agreement with the measurements by Hudda et al. (2014).

It is noteworthy that military airbases in Figure 4g have significantly higher particle number concentrations due to their use of aviation fuel with higher sulfur content but nucleation plumes are not present downwind of these locations (Figure 4j). Particles emitted from military aircraft are represented as primary emissions in the current model calculations. Future measurements should compare particle number concentrations downwind of civilian and military airports to fully evaluate the impact of aviation fuel sulfur content on ambient ultrafine particle concentrations.

Figure 5 illustrates the predicted particle number concentrations associated with primary sources and nucleation in northern California. The relative importance of sources and the prediction of nucleation downwind of major sulfur emissions are consistent in northern and southern California. Natural gas combustion is a notable strong source of ultrafine particles in both regions due to the widespread use of this fuel in numerous residential, commercial, and industrial applications. In many cases, the natural gas combustion particles contribute strongly to the "urban background' concentrations over most California cities without the formation of individual plumes such as those found downwind of LAX. Future measurements could correlate ambient particle number concentrations and natural gas utilization across multiple cities to evaluate whether natural gas combustion is a significant source of particle number concentration.

The spatial patterns of particle number concentrations have been summarized in the main text of the revised manuscript.

[Figure]

Figure 4. Spatial distribution of particle number from major sources in Southern California (unit: kcount/cm3).

[Figure]

Figure 5. Spatial distribution of particle number from major sources in Northern California (unit: kcount/cm3).

The diurnal variation of the natural gas combustion emissions noted by the reviewer were obtained independently from the emissions inventory specified by the California Air Resources Board. The activity pattern is based on energy demand as a function of time of day. Both natural gas combustion and motor vehicle activity follow the diurnal cycle of human activity across California, with peaks in the early morning and late afternoon. The current model predictions suggest that natural gas combustion contributes strongly to this pattern.

We acknowledge that the model predictions match the measured particle trends at some locations but not as well in other locations. We are not claiming that the model is perfect, but we feel that the information available does suggest that natural gas combustion is a major regional source of ultrafine particles that has not been previously recognized.

Reviewer 1 Comment 3: Modeling of growth of ultrafine particles. The approach used to simulate conden- sation/evaporation of sulfuric acid, ammonium, nitric acid, secondary organics on the ultrafine particles in this study is not explained in any detail. There is a rather confusing statement in lines 129-137 that "dynamic condensation/evaporation is not considered". Does this mean that the particles are assumed to be in equilibrium? If yes, how does the model deal with the effect of surface tension on the equilibrium vapor pressure es- pecially in the 10-20 nm range? Do these particles evaporate because their equilibrium vapor pressure is higher than that of the bigger particles? This is a crucial process for the number concentration of the smaller particles and it is not clear that it is simulated properly.

Response: Dynamic simulation of the condensation/evaporation of ultrafine particles is a computationally expensive exercise (Zhang et al. 2004, 2005; Zhang and Wexler 2004). Some of the particles evaporate downwind of sources like freeways, while other particles grow due to the condensation mostly of secondary organic aerosol (Anttila and Kerminen 2003; Troestl et al. 2016). The most extreme changes to the particle size distribution occur within the first few min after emissions to the atmosphere (within 300 m of roadways), with more stable behavior over long time periods.

Regional grid models used to predict regional number concentrations are not well-suited to simulating the dynamic behavior of the near-source particle size distribution for the first few minutes after release to the atmosphere. Evaporation of UFPs near the source is therefore represented by reducing the primary emissions of nano-particles based on measurements conducted at high dilution factors (Xue et al. 2018) or using measurements of particle volatility to estimate the evaporation at high dilution factors (May, Levin, et al. 2013; May, Presto, et al. 2013; Kuwayama et al. 2015). These regionally-representative emissions provide the starting point for the model calculations.

The condensation of fresh sulfate, nitrate, ammonium ion, and SOA onto UFPs with diameters between 10 – 100 nm was simulated using the standard dynamic gas-particle partitioning methods in the model. These calculations do not change the predicted number concentration in the regional atmosphere. Condensation shifts the size distribution upward at a rate of approximately 2-3 nm hr$^{-1}$ under favorable conditions. This has been clarified in the revised manuscript.

Reviewer 2 Comment 1: Definition of particle number concentration. The use of the term particle number concentration throughout this paper is often confusing and sometimes misleading. It is important to always define the lower threshold of the size range of the corresponding concentration. The total particle concentration can be easily a factor of 2 or 3 higher than the concentration of particles with diameter higher than 10 nm (N10).

Response: We will revise the paper to use the term $N_x$ throughout where X refers to the lower size cut of the measurements or model predictions. The term PNC will no longer be used.

Reviewer 2 Comment 2: Growth of freshly nucleated particles to 10 nm. The authors state that they parameterize the growth process following the work of Kerminen and Kulmala (2002). However, this parameterization requires the growth rate (GR) of the particles. The calculation of this rate is non-trivial in a model with coarse aerosol size resolution such as the current one. Errors in the GR can lead to significant errors in the estimation of the contribution of nucleation as a source to particle number. The authors should evaluate the error of this parameterization for their aerosol model.

Response: The growth rate (GR) in the Kerminen and Kulmala (2002) parameterization is one of the factors that accounts for the competition between the condensation and nucleation of over-saturated compounds until the nucleated particles grow to the size of the smallest bin in the regional model at which point this competition is represented explicitly by the model operators. In current study, we predicted the growth of the sulfate particles from nuclei using the equation

$$GR \approx \frac{3 \times 10^{-9}}{\rho_{nuc}} M_{sulf} u_{sulf} C_{sulf} \quad \text{(eq. 1)}$$

following (Kerminen and Kulmala 2002). Here, $\rho nuc$ is the density of the nucleation mode sulfate particles which was set to be 1.77 kg m-3 at 20°C, 1 atm; $M_{sulf}$ is the molecular weight of nucleation mode sulfate particle which was set to be 98 g mol$^{-1}$; Csulf is the vapor concentration of sulfate (H$_2$SO$_4$); and $u_{sulf}$ is temperature ($T$) dependent molecular speed of the sulfate vapor which is calculated as follows, in m s$^{-1}$.

$$u_{sulf} = \sqrt{\frac{8RT}{M_{sulf}}} \quad \text{(eq.2)}$$

According to (Kerminen and Kulmala 2002), uncertainty associated with eq. 1 is minor. Perturbation studies were conducted in the current analysis with a box model configured to represent a single grid cell using the full set of model operators. The GR predicted by eq 1 was multiplied by a factor ranging from 0.5 to 2.0 to test the sensitivity of the model results. Initial conditions were 0.04 ppm O$_3$, 0.05 ppm NO, 0.0 ppm NO$_2$, 0.05 ppm HCHO, 0.1 ppm ISOPRENE, 0.1 ppm BENZENE, and 0.01 ppm ALK5. A nucleation event was initiated at 8am by setting H$_2$SO$_4$ concentrations to 1e7 molecules cm$^{-3}$ and NH$_3$ concentrations to 100 ppt. Figure 6 illustrates the growth of nucleated particles between 5am and 12 noon for July in California. The number concentration of nucleated particles increases to values between 2500 - 3000 #/cm3. SOA condenses on the particles causing their size to increase above 100nm. Coagulation and deposition processes remove particles over time.

Three separate simulations are illustrated in Figure 6 using the nominal GR predicted by eq 1 along with perturbations of 0.5*GR and 2.0*GR. These model perturbations fall almost exactly on top of the basecase simulations, suggesting that results are not overly sensitive to GR during the first few seconds of nuclei growth before calculations are handed off to the regional model algorithms.

[Figure]

Fig 6: Simulated particle nucleation event followed by growth due to SOA condensation under conditions representing July in California. Vertical axis displays the mean diameter of the nuclei mode while color represents the particle number concentration.

Reviewer 2 Comment 3: There is little information provided about the frequency and spatial extent of nucleation in the simulations in the various seasons. This information is needed to under- stand the simulation results.

Response: The concentrations of nucleated particles in August, October, and December are shown in Figure 7 (Southern California) and Figure 8 (Northern California) below. Nucleation events occur in the regions where sulfur emissions are highest (typically airports, shipping ports and refining facilities). Concentrations of nucleated particles are higher in October and December than in August because colder temperatures increase nucleation rates if the precursor $H_2SO_4$ and $NH_3$ concentrations are relatively constant. A significant fraction of the $H_2SO_4$ in the current simulation is produced by the fast conversion of gas-phase $SO_3$ emissions to $H_2SO_4$ in the exhaust plume near the emissions source. $SO_3$ conversion does not depend on the presence of oxidants in the atmosphere and so the higher oxidant concentrations in the summer do not dominate the seasonal nucleation pattern.

Once $H_2SO_4$ forms in the exhaust plumes, it either condenses onto existing particles formed from lower volatility compounds in the plume, or it mixes with $NH_3$ in the background air and nucleates. This process is captured by dilution source sampling measurements that allow for a few minutes of aging time and so the size-resolved emissions profiles for many sources already account for the effects of nucleation within the "near-field" exhaust plume (within a few 10's of meters after emission). $SO_3$ emissions from reciprocating internal combustion engines were therefore set to zero to avoid double counting the new particle formation downwind of these sources in the current study. Regular $SO_2$ emissions from these sources were not modified. Emissions from aircraft jet engines have high exit velocity which promotes rapid mixing with background air. $SO_3$ emissions were left at their nominal levels (3-4% of total SOx) for jet engine aircraft in the current study. The consequence of these model

treatments is that predicted concentrations of nucleated particles are highest downwind of LAX, which agrees with measurements of ambient particle number concentrations (Hudda et al., 2014).

[Figure]

Figure 7: Seasonal variation of nucleated particle concentrations in Southern California.  Units are kcount/cm³.

[Figure]

Figure 8: Seasonal variation of nucleated particle concentrations in Northern California.  Units are kcount/cm³.

Reviewer 2 Comment 4: Emissions from natural gas combustion. A map of the estimated N10 and PM0.1 emissions from this major source is needed (see also comment 1.2). Also the average diurnal profile of the emissions for the domain and the average size distribution should be shown.

Response: The map of emissions from natural gas sources are shown below. Note that particulate matter emissions from all natural gas sources other than reciprocating engines have been reduced by 70% to account for evaporation of particles after emission to the atmosphere (Xue et al. 2018). The average diurnal profile of the natural gas emissions for the domain is shown in Figure S2 and the average size distribution is shown in Figure S3 of the original manuscript.

[Figure]

Figure 5. Daily average natural gas combustion emissions for California.

Reviewer 2 Comment 5: Temporal scale of evaluation. The authors present metrics of the model performance but they do not clarify if these are for hourly, daily, monthly, simulation averages or something else. The text and the corresponding tables do not include this information. Given the availability hourly measurements evaluation at this timescale should be also performed (if it has not been performed yet). The evaluation at a daily scale is also useful.

Response: Comparisons in the manuscript are based on daily averages which corresponds to the shortest averaging time that should be used for the current model results. Comparisons to measurements at hourly and daily time scales are shown in Figure 6 below for particle number

concentration.  The hourly comparisons meet model performance criteria, but have slightly worse performance than the daily averages because the calculations do not fully capture all of the random variability in meteorological patterns and emissions patterns over hourly time scales.  Further work would be required to create accurate model results at hourly time scales, but this effort is beyond the reasonable scope of the current study.  We do not wish to present hourly-average performance metrics in the manuscript because we do not want to encourage the use of the model results at this time scale.

[Figure]

Figure 6 Mean Fractional Bias (MFB) and Mean Fractional Error (MFE) of N10 at 10 sites in California

Reviewer 2 Comment 6: The measurements of particle number refer to N6 and N7 while the predictions to N10. The authors suggest that the average error in the corresponding comparisons should be less than 10

Response: Yes, we feel that comparison between N6 to N10 will only introduce a small amount of uncertainty into the calculation. We would welcome recommendation from the reviewer to adjust the comparison to account for this size difference.

Reviewer 2 Comment 7: The use of qualitative terms (general agreement, agree reasonably well, good agreement) is not helpful and should be avoided.

Response: All qualitative statements will be removed in the final paper.

Reviewer 2 Comment 8: If my understanding of the paper is correct, the current model does not use the dynamic organic aerosol scheme used by Hu et al. (2017). If this is the case, the results regarding the contribution of SOA to PM0.1 in this work should be discussed and should be compared to that version of the model. If it is the same it should be clearly stated.

Response: We apologize that the original text was not clearer. The current study uses the same dynamic organic aerosol scheme used by (Hu et al. 2017). This point will be clarified in the revised manuscript.

Reviewer 2 Comment 9: Contribution of traffic particles. Ronkko et al. (PNAS, 114, 7549-7554, 2017) argued that traffic is an even more important source of particle number, because there are a lot of sub-10 nm particles emitted. Given that the current study does not include primary traffic particles smaller than 10 nm (which of course can grow to larger sizes), can it seriously underestimate the contribution of this source in urban environments?

Response: The measurements in the roadside environment consistently show that traffic dominates nano-particle concentrations. But the measurements moving downwind of the roadside environment show that these traffic nano-particles evaporate and do not increase the urban particle number concentration at distances more than 300 m downwind of the roadway (Zhu, Hinds, Kim, Shen, et al. 2002; Zhu, Hinds, Kim, and Sioutas 2002). This is a measurement conclusion based on independent work not associated with the current manuscript, but it supports the methods used to represent traffic in the regional calculations.

Reviewer 3 Comment 1: The abstract says that simulations have been performed for 2012, 2015 and 2016. However, the presented results are only for 2012. This is important because there are available size distribution measurements for 2015-16 in the modeling domain that can be used for the evaluation of the model predictions (see comment 1.2).

Response: Figures 2 and 3 show the results of the model predictions to CMB results for 2015 and 2016. These findings were added to show that the predictions for $PM_{0.1}$ traffic contributions are in good agreement with measurements, supporting the accuracy of the predictions for the relative importance of traffic vs. other sources of UFP.

The additional particle number concentration measurements in 2015-16 that could be added to the manuscript are the same type as those shown for 2012. All the number count measurements are for sites in the San Francisco Bay Area and Southern California that show essentially the same picture as the plots already included in the manuscript. A separate manuscript is under preparation showing comparisons to all available measurements from 2000-2016. We would like to present the full set of comparisons in a single manuscript rather than further fragmenting this dataset.

Reviewer 3 Comment 2: The predicted correlations between PM2.5 and particle number concentrations can be compared with the corresponding measured correlations as an indirect way to evaluate the model performance.

Response: Table3 below summarizes the predicted correlations between daily-average particle number concentrations and PM2.5 along with the measured correlations for these metrics. Measured correlations ($R^2$) are less than 0.25 at all locations except Santa Rosa where correlations are above 0.5. Model predictions for daily-average particle number concentrations and PM2.5 are more highly correlated, with $R^2$ ranging from 0.22 to 0.73. Locations with high $R^2$ values such as central Los Angeles also have the highest MFB and MFE and so the high correlation between particle number and $PM_{2.5}$ may reflect inaccuracies in the model inputs. At other locations where traditional model performance metrics suggest that predictions are more accurate, the high correlation between particle number and $PM_{2.5}$ may be related to the model grid resolution. The 4km grid resolution used in the calculations smooths the sharp spatial gradients in the ultrafine particle concentration fields (see Response Figure 3). This same issue makes it difficult for point source measurements to accurately represent 4km average number concentrations. The particle number concentrations measured at a fixed monitoring location may not represent the variation in particle number concentrations a few km away. $PM_{2.5}$ concentration gradients are smoother, making model predictions and point measurements easier to compare. This analysis suggests that the model results contained in the current manuscript identify several important sources of ultrafine particles, but more work will be required to fully evaluate these results and possibly further refine the population exposure calculations.

Table 3. Daily-average correlation ($R^2$) between PM2.5 mass and particle number concentration at 8 sites in California.

| $R^2$ | Livermore | Redwood City | San Pablo | Santa Rosa | Anaheim | Central LA | Compton | Rubidoux |
|---|---|---|---|---|---|---|---|---|
| Obs | 0.04 | 0.01 | 0.16 | 0.58 | 0.08 | 0.14 | 0.15 | 0.22 |
| Sim | 0.28 | 0.49 | 0.55 | 0.22 | 0.51 | 0.73 | 0.61 | 0.50 |

Reviewer 3 Comment 3: Lines 66-67 "when nucleation algorithms were not standardized". This statement is confusing.

Response: Will be changed to "…when different nucleation algorithms were used".

Reviewer 3 Comment 4: Are the sulfate and nitrate concentrations shown in Table S4 for PM2.5 or for another size range?

Response: PM2.5

Reviewer 3 Comment 5: Table 1 should probably also include the predicted and measured average number concentrations.

Response: Revised Table 1 shown below.

|  | Ave Obs. Particles cm-3 | Ave Sim. Particles cm-3 | MFB | MFE | RMSE Particles cm-3 |
|---|---|---|---|---|---|
| Livermore | 8219 | 9201 | 0.10 | 0.09 | 3615 |
| Redwood city | 11500 | 11325 | 0.02 | 0.08 | 1132 |
| San Pablo | 10481 | 15822 | 0.30 | 0.31 | 10302 |
| Santa Rosa | 8655 | 8967 | 0.05 | 0.15 | 2063 |
| Anaheim | 12850 | 14812 | 0.12 | 0.14 | 4239 |
| Central LA | 17378 | 25376 | 0.37 | 0.38 | 10328 |
| Compton | 16203 | 21036 | 0.24 | 0.26 | 8127 |
| Huntington | 23207 | 24103 | 0.04 | 0.08 | 3698 |
| Inland-Valley | 15028 | 16875 | 0.12 | 0.17 | 4290 |
| Rubidoux | 10728 | 11920 | 0.11 | 0.16 | 3069 |

Reviewer 3 Comment 6: The terms "measured" and "predicted" should be used everywhere in Section 3.2.1 and other parts of the paper in which predictions are compared to measurements.

Response: This change will be made as suggested to the degree possible, but the term "measured" is too simplistic. The molecular marker measurements feed into a model prediction using the Chemical Mass Balance (CMB) model that has many model inputs and assumptions. There are no direct measurements of source contributions to PM0.1 – just model predictions using different techniques.

Reviewer 3 Comment 7: The number of samples and their duration corresponding to the results of Figs. 2-3 should be stated in the caption.

Response: Monthly average samples constructed from 3-day average measurements. This information will be added to figure caption as requested.

Reviewer 3 Comment 8: Line 296. Figures 4-6 and 7-9 do not show the seasonal variation of the corresponding variables. They show data (are these daily averages or something else) for different days in different seasons. These figures could be improved if they were split in four parts for the different periods simulated. The discussion could also be improved if the actual seasonal averages were shown (may be in the SI) and discussed.

Response: Figures show daily variation over months that span multiple seasons. The x-axis on each Figure will be improved to show the months more clearly. Figure captions will expanded to better explain the results.

Reviewer 3 Comment 9: Figure S2. What is A, B, and C? What is the average pattern in the domain?

Response: A, B and C represent different diurnal profiles used for different natural gas sources or regions based on information supplied by the California Air Resources Board. The average diurnal profile will be added to Figure S2.

[Figure]

Figure S2 Diurnal profiles of no-residential natural gas emissions. A, B and C represents different types of diurnal profiles applied to natural gas emissions in the model. Black curve represents the average pattern in the domain.

References

Hudda, N., Gould, T., Hartin, K., Larson, T. V., and Fruin, S. A.: Emissions from an International Airport Increase Particle Number Concentrations 4-fold at 10 km Downwind, Environmental science & technology, 48, 6628-6635, 10.1021/es5001566, 2014.
Zhang, K. M., Wexler, A. S., Zhu, Y. F., Hinds, W. C., and Sioutas, C.: Evolution of particle number distribution near roadways. Part II: the 'road-to-ambient' process, Atmospheric Environment, 38, 6655-6665, 10.1016/j.atmosenv.2004.06.044, 2004.
Zhang, K. M., Wexler, A. S., Niemeier, D. A., Zhu, Y. F., Hinds, W. C., and Sioutas, C.: Evolution of particle number distribution near roadways. Part III: Traffic analysis and on-road size resolved particulate emission factors, Atmospheric Environment, 39, 4155-4166, 10.1016/j.atmosenv.2005.04.003, 2005.
Zhu, Y. F., Hinds, W. C., Kim, S., Shen, S., and Sioutas, C.: Study of ultrafine particles near a major highway with heavy-duty diesel traffic, Atmospheric Environment, 36, 4323-4335, 10.1016/s1352-2310(02)00354-0, 2002a.
Zhu, Y. F., Hinds, W. C., Kim, S., and Sioutas, C.: Concentration and size distribution of ultrafine particles near a major highway, Journal of the Air & Waste Management Association, 52, 1032-1042, 10.1080/10473289.2002.10470842, 2002b.

---

## Author Response (AR2)

Reviewer 1 Comment 1.1: Sources of particle number. The authors calculate the particle number contributed by each source based on the corresponding mass (equation 1 in the paper). This is wrong for two reasons. First a significant fraction of the particle mass is secondary (sulfates, nitrates, secondary organic aerosol). When the secondary mass increases, the contribution of the corresponding source to particle number does not. Second, co- agulation involves particles from different sources. It is not clear to which source the authors assign the particle resulting from the coagulation of two particles from different sources. Both of these problems are quite important for ultrafine particle number con- centrations. The errors of this oversimplified approach should be estimated (at least for one period) with careful zero-out analysis (e.g., removing only the ultrafines and not the larger particles to avoid changes in the condensation and coagulation sinks). If the error is significant the corresponding part of the work should be redone or should be replaced with just a description of the contributions to emissions for different size ranges.

Response: We apologize that the methods to calculate particle number were not explained clearly in the first version of the paper. The text on lines 310-325 has been updated to provide more details. The model framework uses a moving sectional approach to conserve particle number and mass while letting particle radius increase due to condensation (Kleeman, Cass, and Eldering 1997). The method to calculate source contributions to number concentration is performed for each moving section individually. Number is explicitly conserved and correctly apportioned to sources in this algorithm.

Each particle source type / moving size bin includes an artificial tracer equal to 1% of the primary particle mass. The mass of this tracer is related to the number of particles by the equation

Tracer_source_i * 100 = N_source_i * 3.14159/6 * Dp_bin * density_source_i

This equation can be easily rearranged to solve for N_source_i as a function of Tracer_source_i in each size bin. Again, since the model uses a moving sectional approach, number and tracer mass are exactly conserved. Condensation/evaporation changes the particle diameter as semi-volatile components move on and off the particle but this does not change Tracer_source_i or N_source_i. The moving sectional approach greatly simplifies the source apportionment of particle number compared to other models that use fixed particle size bins with condensation / evaporation transferring material between bins.

The text on lines 326-337 has been updated to describe how source apportionment calculations handle coagulation. Coagulation is fastest between very small particles and relatively large particles in the atmosphere. The net effect of coagulation is to remove ultrafine particles from the atmosphere as they collide and join the particles larger than 100 nm. This loss mechanism is accurately simulated in the model calculations. The rate of "self-coagulation" between two ultrafine particles that produces a particle still in the ultrafine particle size range is negligible at atmospherically relevant concentrations. Table 2 compares the timescale for 0.01 μm particles coagulating with other 0.01 μm particles and coagulating with 0.1 μm particles based on size distributions measured in a typical suburban environment in a California city (see Figure 1). The coagulation of timescale between two 0.01 μm particles is 209 hrs (8.7 days) while the coagulation timescale between 0.01 μm particles and 0.1 μm particles is 4.4 hours. Therefore, self-coagulation between ultrafine particles smaller than 60 nm is much less significant than coagulation between ultrafine particles and larger particles (acting as a loss mechanism for ultrafine particles in the atmosphere).

Table 2 Time scale for coagulation between 0.01 µm particles with 0.01 µm particles and 0.1 µm particles in a typical suburban environment in California.  See Fig 1 for size distribution used for calculations.

| Particle size | 0.01 | 0.1 |
|---|---|---|
| Coagulation Coefficient(cm³/s) | 1.90E-09 | 2.50E-08 |
| PM Number concentration (#/cm³) | 1.40E+03 | 5.00E+03 |
| timescale (hours) | 208.9 | 4.4 |

[Figure]

Figure 1 Particle size distribution measured in a typical suburban environment in a California city

Source apportionment calculations treat coagulation events between very small particles and very large particles in a manner analogous to condensation.  When two particles coagulate, the mass of the smaller particle is added to the mass of the larger particle.  The number concentration of the smaller particle is discarded while the number concentration of the larger particle stays constant. This slightly reduces the accuracy of source apportionment calculations for particle number in the larger size bins because the Tracer_source mass in the larger size bin is no longer proportional to the number concentration from that source.  This issue is relatively minor since size bins larger than 1µm that act as the dominant sink during particle coagulation events typically account for less than 5% of the total number concentration.

Perturbation studies were conducted as requested by the reviewer by setting the UFP emissions for on-road gasoline vehicles to zero during the month August 2012.  Emissions of gases and emissions of larger particles from on-road vehicles were not changed.  The difference between this perturbation simulation vs. the basecase simulation was calculated to estimate the number concentration of particles associated with on-road gasoline vehicles.  This "zero-out" concentration is then compared to the standard model source-apportionment calculations in Figure 2 below (shown as Figure 3 in the revised manuscript).  The two methods for number source apportionment yield very similar spatial patterns and very similar maximum concentrations of approximately 0.5 kcounts/cm³.  The tracer source apportionment method

accounts for all particle sizes which produces slightly higher concentrations than the zero-out method that only considered particles smaller than 100 nm.  This test confirms that the online source apportionment methods for number in the current study work correctly.

[Figure]

Figure 2 Particle number concentrations associated with on-road gasoline vehicles calculated using the zero-out method and the artificial tracer method in August 2012.

Reviewer 1 Comment 1.2: Importance of non-residential natural gas combustion as a source of ultrafine particles.  This is clearly the most important, but also the most controversial finding of the study. The evidence provided to support this potentially very important result is rather weak and the authors miss a lot of opportunities to strengthen their argument.

The first is the use of size distributions. The predicted size distributions from this source apparently peak in the 10-20 nm size range. There are a lot of available size distribution measurements in the area that can be directly compared with the model predictions.    My understanding however is that the measured number size distributions (not immediately next to freeways) peak at the 35-40 nm range (see for example Sowlat et al., 2016). Some of these size distribution measurements are available for the periods that have been simulated so a comparison of size distributions (including sources) could be performed without much effort.

The second is the use of the spatial distribution of particle number. The predicted concentration maps are not shown, but one would expect much higher concentrations near the corresponding major source areas. Traffic should have quite a different spatial pattern. There have been also a lot of particle number distribution measurements in California during the last decade. An effort to test if the predicted patterns match the observed ones would help.

The third is the average diurnal variation. However, this study assumes that the non- residential natural gas emissions have a similar temporal pattern as traffic (Figure S2). So the observed rush-hour peak in particle number that all previous studies assign       to traffic, here is explained by natural gas combustion. However, more careful spatio- temporal analysis could help strengthen (or weaken) the conclusion. For example, the predicted morning number peak in Rubidoux in summer does not exist in the measurements. The situation is even worse in midday during the winter suggesting that emissions from this source are clearly overestimated in this area. Is this helpful? Is this area dominated by these emissions or is the sampling site an exception? On the other hand, the model performs well in other areas so one could make the opposite argument site by site.  However,  without  using  all  the information  about  predicted  patterns  in space and time it is difficult to reach a conclusion.

Response: Xue et al. showed that the primary size distribution for natural gas combustion peaks at approximately 20 nm but the size mode grows to approximately 60 nm after 3 hrs of aging in a smog chamber with a representative urban atmosphere consistent of realistic concentrations of VOCs and NOx under realistic UV intensity (Xue et al. 2018).  Similar growth occurs in model calculations meaning that the natural gas particles do not stay static at 20 nm in the atmosphere. This point has been clarified on lines 490-491 and in Figure 1 of the revised manuscript. The measurements of larger particles in the atmosphere therefore do not definitively identify sources.   Expert opinion is still required to interpret the size distributions and assign them to sources.  The results of the current study should help refine those expert opinions in the future.

Many of the spatial patterns measured for airborne particle number concentrations have focused on the gradients around roads (see for example (Zhang et al., 2005;Zhang et al., 2004;Zhu et al., 2002a;Zhu et al., 2002b)).  Likewise, the study performed by Solwat et al. (2016) referenced by the reviewer was carried out within 150m of a major freeway and so the reported particle size distributions are dominated by traffic sources.  These gradients are impossible to resolve using a regional model with 4km resolution.  A limited set of additional simulations were conducted using the WRF/Chem model configured with Large Eddy Simulation (LES) around Oakland California so that spatial scales down to 250m could be examined.  Maps of the predicted ultrafine particle mass concentrations for gasoline, diesel, food cooking, wood combustion, and natural gas combustion particles are shown in Figure 3 below (shown as Figure 4 in the revised manuscript).  At 250m resolution, ultrafine particles from diesel engines peak on major transportation corridors while ultrafine particles from gasoline vehicles are more diffuse reflecting their increased activity on adjacent surface streets.  Ultrafine particles from natural gas combustion are even more diffuse reflecting contributions from area sources across the region.  As the spatial resolution decreases to 1km and then 4km, the fine details around roadways are artificially diluted in the larger grid cells.  This process shifts the dominant source of ultrafine particles over roadways from diesel engines at 250m resolution to natural gas combustion at 4km resolution.  This discussion is now included on lines 363-377 of the revised manuscript.

The ultrafine particle model simulations summarized in Figure 3 are consistent with measurements of particle number in the proximity of roadways which show that the traffic contribution to particle number concentration decays to background levels within 300 m (Zhu, Hinds, Kim, Shen, et al. 2002; Zhu, Hinds, Kim, and Sioutas 2002). The measurements made by Zhu et al. indicate that the traffic contribution to regional number concentration cannot be distinguished from other sources on a regional scale using 4km grid cells which is the focus of this study.

Repeating all of the simulations at 250m resolution is beyond the scope of the current study.  We emphasize the regional scope of the simulations in the main text of the revised manuscript to inform the readers about the appropriate interpretation of the current results, and we have also added "regional" to the title of the manuscript.

[Figure]

Figure 3: PM$_{0.1}$ mass concentration associated with on-road diesel, on-road gasoline, and natural gas combustion at 250m, 1km, and 4km resolution over Oakland, California.

Hudda et al. (2014) found that particle number concentrations increased by a factor of four to eight downwind of the Los Angeles International Airport (LAX) based on measurements in June-July 2013. Total ground-level number concentrations in the LAX plume reached 60-70 *$10^3$ counts/$cm^3$. Figure 4 illustrates the predicted number concentration associated with primary emissions (Figures 4a-i) and nucleation (Figure j) averaged over the months Aug-Dec 2012. Figure 4g shows that primary aircraft emissions in the LAX plume are predicted to account for 8 * $10^3$ counts/$cm^3$ and Figure 4j shows that nucleation of aircraft emissions in the LAX plume are predicted to account for 45 * $10^3$ counts/$cm^3$ yielding a total number concentration associated with LAX aircraft of approximately 53 * $10^3$ counts/$cm^3$. Given the 4km spatial resolution of the model calculations, these findings are in good agreement with the measurements by Hudda et al. (2014).

It is noteworthy that military airbases in Figure 4g have significantly higher particle number concentrations due to their use of aviation fuel with higher sulfur content but nucleation plumes are not present downwind of these locations (Figure 4j). Particles emitted from military aircraft are represented as primary emissions in the current model calculations. Future measurements should compare particle number concentrations downwind of civilian and military airports to fully evaluate the impact of aviation fuel sulfur content on ambient ultrafine particle concentrations.

The discussion of the spatial patterns of number concentrations downwind of airports is summarized in Figure 18 and on lines 559-576 of the revised manuscript.

Figure 5 illustrates the predicted particle number concentrations associated with primary sources and nucleation in northern California. The relative importance of sources and the prediction of nucleation downwind of major sulfur emissions are consistent in northern and southern California. Natural gas combustion is a notable strong source of ultrafine particles in both regions due to the widespread use of this fuel in numerous residential, commercial, and industrial applications. In many cases, the natural gas combustion particles contribute strongly to the "urban background' concentrations over most California cities without the formation of individual plumes such as those found downwind of LAX. Future measurements could correlate ambient particle number concentrations and natural gas utilization across multiple cities to evaluate whether natural gas combustion is a significant source of particle number concentration.

The spatial patterns of nucleated particle number concentrations have been summarized on lines 595-620 in the main text of the revised manuscript along with Figures 20 and 21.

[Figure]

Figure 4. Spatial distribution of particle number from major sources in Southern California (unit: kcount/cm3).

[Figure]

Figure 5. Spatial distribution of particle number from major sources in Northern California (unit: kcount/cm3).

Lines 524-526 have been updated to clarify that the diurnal variation of the natural gas combustion emissions noted by the reviewer were obtained independently from the emissions inventory specified by the California Air Resources Board. The activity pattern is based on energy demand as a function of time of day. Both natural gas combustion and motor vehicle activity follow the diurnal cycle of human activity across California, with peaks in the early morning and late afternoon. The current model predictions suggest that natural gas combustion contributes strongly to this pattern.

We acknowledge that the model predictions match the measured particle trends at some locations but not as well in other locations. We are not claiming that the model is perfect, but we feel that the information available does suggest that natural gas combustion is a major regional source of ultrafine particles that has not been previously recognized.

Reviewer 1 Comment 1.3: Modeling of growth of ultrafine particles. The approach used to simulate conden- sation/evaporation of sulfuric acid, ammonium, nitric acid, secondary organics on the ultrafine particles in this study is not explained in any detail. There is a rather confusing statement in lines 129-137 that "dynamic condensation/evaporation is not considered". Does this mean that the particles are assumed to be in equilibrium? If yes, how does the model deal with the effect of surface tension on the equilibrium vapor pressure es- pecially in the 10-20 nm range? Do these particles evaporate because their equilibrium vapor pressure is higher than that of the bigger particles? This is a crucial process for the number concentration of the smaller particles and it is not clear that it is simulated properly.

Response: Dynamic simulation of the condensation/evaporation of ultrafine particles is a computationally expensive exercise (Zhang et al. 2004, 2005; Zhang and Wexler 2004). Some of the particles evaporate downwind of sources like freeways, while other particles grow due to the condensation mostly of secondary organic aerosol (Anttila and Kerminen 2003; Troestl et al. 2016). The most extreme changes to the particle size distribution occur within the first few min after emissions to the atmosphere (within 300 m of roadways), with more stable behavior over long time periods.

Regional grid models used to predict regional number concentrations are not well-suited to simulating the dynamic behavior of the near-source particle size distribution for the first few minutes after release to the atmosphere. Evaporation of UFPs near the source is therefore represented by reducing the primary emissions of nano-particles based on measurements conducted at high dilution factors (Xue et al. 2018) or using measurements of particle volatility to estimate the evaporation at high dilution factors (May, Levin, et al. 2013; May, Presto, et al. 2013; Kuwayama et al. 2015). These regionally-representative emissions provide the starting point for the model calculations.

These points have been clarified on lines 159-175 of the updated manuscript.

The condensation of fresh sulfate, nitrate, ammonium ion, and SOA onto UFPs with diameters between 10 – 100 nm was simulated using the standard dynamic gas-particle partitioning methods in the model. These calculations do not change the predicted number concentration in the regional atmosphere. Condensation shifts the size distribution upward at a rate of approximately 2-3 nm hr$^{-1}$ under favorable conditions. This has been clarified on lines 158-159 in the revised manuscript.

Reviewer 1 Comment 2.1: Definition of particle number concentration. The use of the term particle number concentration throughout this paper is often confusing and sometimes misleading. It is important to always define the lower threshold of the size range of the corresponding concentration. The total particle concentration can be easily a factor of 2 or 3 higher than the concentration of particles with diameter higher than 10 nm (N10).

Response: We will revise the paper to use the term $N_x$ throughout where X refers to the lower size cut of the measurements or model predictions. The term PNC will no longer be used.

Reviewer 1 Comment 2.2: Growth of freshly nucleated particles to 10 nm. The authors state that they parameterize the growth process following the work of Kerminen and Kulmala (2002). However, this parameterization requires the growth rate (GR) of the particles. The calculation of this rate is non-trivial in a model with coarse aerosol size resolution such as the current one. Errors in the GR can lead to significant errors in the estimation of the contribution of nucleation as a source to particle number. The authors should evaluate the error of this parameterization for their aerosol model.

Response: The text on lines 104-129 and Figure 1 in the revised manuscript have been added to address this point. The growth rate (GR) in the Kerminen and Kulmala (2002) parameterization is one of the factors that accounts for the competition between the condensation and nucleation of over-saturated compounds until the nucleated particles grow to the size of the smallest bin in the regional model at which point this competition is represented explicitly by the model operators. In current study, we predicted the growth of the sulfate particles from nuclei using the equation

$$GR \approx \frac{3 \times 10^{-9}}{\rho_{nuc}} M_{sulf} u_{sulf} C_{sulf} \quad \text{(eq. 1)}$$

following (Kerminen and Kulmala 2002). Here, $\rho nuc$ is the density of the nucleation mode sulfate particles which was set to be 1.77 kg m-3 at 20ºC, 1 atm; $M_{sulf}$ is the molecular weight of nucleation mode sulfate particle which was set to be 98 g mol$^{-1}$; Csulf is the vapor concentration of sulfate ($H_2SO_4$); and $u_{sulf}$ is temperature ($T$) dependent molecular speed of the sulfate vapor which is calculated as follows, in m s$^{-1}$.

$$u_{sulf} = \sqrt{\frac{8RT}{M_{sulf}}} \quad \text{(eq.2)}$$

According to (Kerminen and Kulmala 2002), uncertainty associated with eq. 1 is minor. Perturbation studies were conducted in the current analysis with a box model configured to represent a single grid cell using the full set of model operators. The GR predicted by eq 1 was multiplied by a factor ranging from 0.5 to 2.0 to test the sensitivity of the model results. Initial conditions were 0.04 ppm $O_3$, 0.05 ppm NO, 0.0 ppm $NO_2$, 0.05 ppm HCHO, 0.1 ppm ISOPRENE, 0.1 ppm BENZENE, and 0.01 ppm ALK5. A nucleation event was initiated at 8am by setting $H_2SO_4$ concentrations to 1e7 molecules cm$^{-3}$ and $NH_3$ concentrations to 100 ppt. Figure 6 illustrates the growth of nucleated particles between 5am and 12 noon for July in California. The number concentration of nucleated particles increases to values between 2500 - 3000 #/cm3. SOA condenses on the particles causing their size to increase above 100nm. Coagulation and deposition processes remove particles over time.

Three separate simulations are illustrated in Figure 6 using the nominal GR predicted by eq 1 along with perturbations of 0.5*GR and 2.0*GR. These model perturbations fall almost exactly on top of the basecase simulations, suggesting that results are not overly sensitive to GR during the first few seconds of nuclei growth before calculations are handed off to the regional model algorithms.

[Figure]

Fig 6: Simulated particle nucleation event followed by growth due to SOA condensation under conditions representing July in California. Vertical axis displays the mean diameter of the nuclei mode while color represents the particle number concentration.

Reviewer 1 Comment 2.3: There is little information provided about the frequency and spatial extent of nucleation in the simulations in the various seasons. This information is needed to under- stand the simulation results.

Response: The text on lines 595-620 in the revised manuscript have been added to address this point and the discussion is repeated below. The concentrations of nucleated particles in August, October, and December are shown in Figure 7 (Southern California) and Figure 8 (Northern California) below. Nucleation events occur in the regions where sulfur emissions are highest (typically airports, shipping ports and refining facilities). Concentrations of nucleated particles are higher in October and December than in August because colder temperatures increase nucleation rates if the precursor $H_2SO_4$ and $NH_3$ concentrations are relatively constant. A significant fraction of the $H_2SO_4$ in the current simulation is produced by the fast conversion of gas-phase $SO_3$ emissions to $H_2SO_4$ in the exhaust plume near the emissions source. $SO_3$ conversion does not depend on the presence of oxidants in the atmosphere and so the higher oxidant concentrations in the summer do not dominate the seasonal nucleation pattern.

Once $H_2SO_4$ forms in the exhaust plumes, it either condenses onto existing particles formed from lower volatility compounds in the plume, or it mixes with $NH_3$ in the background air and nucleates. This process is captured by dilution source sampling measurements that allow for a few minutes of aging time and so the size-resolved emissions profiles for many sources already account for the effects of nucleation within the "near-field" exhaust plume (within a few 10's of meters after emission). $SO_3$ emissions from reciprocating internal combustion engines were therefore set to zero to avoid double

counting the new particle formation downwind of these sources in the current study. Regular SO₂ emissions from these sources were not modified. Emissions from aircraft jet engines have high exit velocity which promotes rapid mixing with background air. SO₃ emissions were left at their nominal levels (3-4% of total SOx) for jet engine aircraft in the current study. The consequence of these model treatments is that predicted concentrations of nucleated particles are highest downwind of LAX, which agrees with measurements of ambient particle number concentrations (Hudda et al., 2014).

[Figure]

Figure 7: Seasonal variation of nucleated particle concentrations in Southern California. Units are kcount/cm³.

[Figure]

Figure 8: Seasonal variation of nucleated particle concentrations in Northern California. Units are kcount/cm³.

Reviewer 1 Comment 2.4: Emissions from natural gas combustion. A map of the estimated N10 and PM0.1 emissions from this major source is needed (see also comment 1.2). Also the average diurnal profile of the emissions for the domain and the average size distribution should be shown.

Response: The map of emissions from natural gas sources are shown below and now included in the SI along with the diurnal profile of the natural gas emissions. Note that particulate matter emissions from all natural gas sources other than reciprocating engines have been reduced by 70% to account for evaporation of particles after emission to the atmosphere (Xue et al. 2018). The average diurnal profile of the natural gas emissions for the domain is shown in Figure S2 and the average size distribution is shown in Figure S3 of the original manuscript.

[Figure]

Figure 5. Daily average natural gas combustion emissions for California.

Reviewer 1 Comment 2.5: Temporal scale of evaluation. The authors present metrics of the model performance but they do not clarify if these are for hourly, daily, monthly, simulation averages or something else. The text and the corresponding tables do not include this information. Given the availability hourly measurements evaluation at this timescale should be also performed (if it has not been performed yet). The evaluation at a daily scale is also useful.

Response: Comparisons in the manuscript are based on daily averages which corresponds to the shortest averaging time that should be used for the current model results. Comparisons to measurements at hourly and daily time scales are shown in Figure 6 below for particle number concentration. The hourly comparisons meet model performance criteria, but have slightly worse performance than the daily averages because the calculations do not fully capture all of the random variability in meteorological patterns and emissions patterns over hourly time scales. Further work would be required to create accurate model results at hourly time scales, but this effort is beyond the reasonable scope of the current study. We do not wish to present hourly-average performance metrics in the manuscript because we do not want to encourage the use of the model results at this time scale.

[Figure]

Figure 6 Mean Fractional Bias (MFB) and Mean Fractional Error (MFE) of "best-fit" N10 at 10 sites in California

Reviewer 1 Comment 2.6: The measurements of particle number refer to N6 and N7 while the predictions to N10. The authors suggest that the average error in the corresponding comparisons should be less than 10

Response: Yes, we feel that comparison between N6 to N10 will only introduce a small amount of uncertainty into the calculation. We would welcome recommendation from the reviewer to adjust the comparison to account for this size difference.

Reviewer 1 Comment 2.7: The use of qualitative terms (general agreement, agree reasonably well, good agreement) is not helpful and should be avoided.

Response: All qualitative statements have been removed in the final paper.

Reviewer 1 Comment 2.8: If my understanding of the paper is correct, the current model does not use the dynamic organic aerosol scheme used by Hu et al. (2017). If this is the case, the results regarding the contribution of SOA to PM0.1 in this work should be discussed and should be compared to that version of the model. If it is the same it should be clearly stated.

Response: We apologize that the original text was not clearer. The current study uses the same dynamic organic aerosol scheme used by (Hu et al. 2017). This point has been clarified on lines 158-159 in the revised manuscript.

Reviewer 1 Comment 2.9: Contribution of traffic particles. Ronkko et al. (PNAS, 114, 7549-7554, 2017) argued that traffic is an even more important source of particle number, because there are a lot of sub-10 nm particles emitted. Given that the current study does not include primary traffic particles smaller than 10 nm (which of course can grow to larger sizes), can it seriously underestimate the contribution of this source in urban environments?

Response: The measurements in the roadside environment consistently show that traffic dominates nano-particle concentrations. But the measurements moving downwind of the roadside environment show that these traffic nano-particles evaporate and do not increase the urban particle number concentration at distances more than 300 m downwind of the roadway (Zhu, Hinds, Kim, Shen, et al. 2002; Zhu, Hinds, Kim, and Sioutas 2002). This is a measurement conclusion based on independent work not associated with the current manuscript, but it supports the methods used to represent traffic in the regional calculations. These points have been clarified on lines 371-374 in the revised manuscript.

Reviewer 1 Comment 3.1: The abstract says that simulations have been performed for 2012, 2015 and 2016. However, the presented results are only for 2012. This is important because there are available size distribution measurements for 2015-16 in the modeling domain that can   be used for the evaluation of the model predictions (see comment 1.2).

Response: Figures 2 and 3 show the results of the model predictions to CMB results for 2015 and 2016. These findings were added to show that the predictions for $PM_{0.1}$ traffic contributions are in good agreement with measurements, supporting the accuracy of the predictions for the relative importance of traffic vs. other sources of UFP.

The additional particle number concentration measurements in 2015-16 that could be added to the manuscript are the same type as those shown for 2012. All the number count measurements are for sites in the San Francisco Bay Area and Southern California that show essentially the same picture as the plots already included in the manuscript. A separate manuscript is under preparation showing comparisons to all available measurements from 2000-2016. We would like to present the full set of comparisons in a single manuscript rather than further fragmenting this dataset.

Reviewer 1 Comment 3.2: The predicted correlations between PM2.5 and particle number concentrations    can be compared with the corresponding measured correlations as an indirect way to evaluate the model performance.

Response: Table3 below summarizes the predicted correlations between daily-average particle number concentrations and PM2.5 along with the measured correlations for these metrics. Measured correlations ($R^2$) are less than 0.25 at all locations except Santa Rosa where correlations are above 0.5. Model predictions for daily-average particle number concentrations and PM2.5 are more highly

correlated, with $R^2$ ranging from 0.22 to 0.73.  Locations with high $R^2$ values such as central Los Angeles also have the highest MFB and MFE and so the high correlation between particle number and $PM_{2.5}$ may reflect inaccuracies in the model inputs.  At other locations where traditional model performance metrics suggest that predictions are more accurate, the high correlation between particle number and $PM_{2.5}$ may be related to the model grid resolution.   The 4km grid resolution used in the calculations smooths the sharp spatial gradients in the ultrafine particle concentration fields (see Response Figure 3). This same issue makes it difficult for point source measurements to accurately represent 4km average number concentrations.  The particle number concentrations measured at a fixed monitoring location may not represent the variation in particle number concentrations a few km away. $PM_{2.5}$ concentration gradients are smoother, making model predictions and point measurements easier to compare. This analysis suggests that the model results contained in the current manuscript identify several important sources of ultrafine particles, but more work will be required to fully evaluate these results and possibly further refine the population exposure calculations.

Table 3. Daily-average correlation ($R^2$) between PM2.5 mass and particle number concentration at 8 sites in California.

| $R^2$ | Livermore | Redwood City | San Pablo | Santa Rosa | Anaheim | Central LA | Compton | Rubidoux |
|---|---|---|---|---|---|---|---|---|
| Obs | 0.04 | 0.01 | 0.16 | 0.58 | 0.08 | 0.14 | 0.15 | 0.22 |
| Sim | 0.28 | 0.49 | 0.55 | 0.22 | 0.51 | 0.73 | 0.61 | 0.50 |

The points above have been added at lines 289-305 along with Table 2 in the revised manuscript.

Reviewer 1 Comment 3.3: Lines 66-67 "when nucleation algorithms were not standardized". This statement is confusing.

Response: Will be changed to "…when different nucleation algorithms were used".

Reviewer 1 Comment 3.4: Are the sulfate and nitrate concentrations shown in Table S4 for PM2.5 or for another size range?

Response: PM2.5

Reviewer 1 Comment 3.5: Table 1 should probably also include the predicted and measured average number concentrations.

Response: Revised Table 1 shown below and in revised manuscript.

| | Ave Obs. Particles cm-3 | Ave Sim. Particles cm-3 | MFB | MFE | RMSE Particles cm-3 |
|---|---|---|---|---|---|

| | | | | | |
|---|---|---|---|---|---|
| Livermore | 8219 | 9201 | 0.10 | 0.09 | 3615 |
| Redwood city | 11500 | 11325 | 0.02 | 0.08 | 1132 |
| San Pablo | 10481 | 15822 | 0.30 | 0.31 | 10302 |
| Santa Rosa | 8655 | 8967 | 0.05 | 0.15 | 2063 |
| Anaheim | 12850 | 14812 | 0.12 | 0.14 | 4239 |
| Central LA | 17378 | 25376 | 0.37 | 0.38 | 10328 |
| Compton | 16203 | 21036 | 0.24 | 0.26 | 8127 |
| Huntington | 23207 | 24103 | 0.04 | 0.08 | 3698 |
| Inland-Valley | 15028 | 16875 | 0.12 | 0.17 | 4290 |
| Rubidoux | 10728 | 11920 | 0.11 | 0.16 | 3069 |

Reviewer 1 Comment 3.6: The terms "measured" and "predicted" should be used everywhere in Section 3.2.1 and other parts of the paper in which predictions are compared to measurements.

Response: This change will be made as suggested to the degree possible, but the term "measured" is too simplistic. The molecular marker measurements feed into a model prediction using the Chemical Mass Balance (CMB) model that has many model inputs and assumptions. There are no direct measurements of source contributions to PM0.1 – just model predictions using different techniques.

Reviewer 1 Comment 3.7: The number of samples and their duration corresponding to the results of Figs. 2-3 should be stated in the caption.

Response: Monthly average samples constructed from 3-day average measurements. This information has been added to figure caption as requested.

Reviewer 1 Comment 3.8: Line 296. Figures 4-6 and 7-9 do not show the seasonal variation of the corresponding variables. They show data (are these daily averages or something else) for different days in different seasons. These figures could be improved if they were split in four parts for the different periods simulated. The discussion could also be improved if the actual seasonal averages were shown (may be in the SI) and discussed.

Response: Figures show daily variation over months that span multiple seasons. The x-axis on each Figure has been improved to show the months more clearly. Figure captions have been expanded to better explain the results.

Reviewer 1 Comment 3.9: Figure S2. What is A, B, and C? What is the average pattern in the domain?

Response: A, B and C represent different diurnal profiles used for different natural gas sources or regions based on information supplied by the California Air Resources Board. The average diurnal profile has been added to Figure S2.

[Figure]

Figure S2 Diurnal profiles of no-residential natural gas emissions. A, B and C represents different types of diurnal profiles applied to natural gas emissions in the model. Black curve represents the average pattern in the domain.

Reviewer 2 Comments:

Reviewer 2 Comment 1.Maybe MFE and MFB are very useful to present the model performance. But I would suggest the authors also provide the correlation between the predicted and measured results, which is more straightforward.

Response: Pearson correlation coefficients (R) have been added to revised Table 1.

Reviewer 2 Comment 2.Page 8, line 230: I am confused why the author raise the value of 8% (N7-10/N7-1000) here? Did you use it to correct model results? If so, then this value is measured at Fresno supersite, which is located near roadways with moderate traffic. So could it be used for all the cases? Also, the particle number concentration has a significant diurnal variation, especially during the nucleation event days. But the authors only compare the daily average, this might be problematic. Concerning the particle number simulation, the number size distribution is also very important. Do they have any number size distribution measurements on the sites? I think it might be worth to compare.

Response: The value of 8% is provided to give the reader some idea about the uncertainty introduced by comparing slightly different particle size fractions.  No adjustments were made to either the measurements or model predictions.  This is stated in the revised manuscript at line 294:

"Previous studies conducted at Fresno, California, suggest that $N_{7-10}$ accounts for approximately 8% of $N_7$ (Watson et al., 2011), and so some amount of negative bias is expected when comparing predicted $N_{10}$ to measured $N_7$".

Comparisons in the manuscript are based on daily averages, which corresponds to the shortest averaging time that should be used for the current model results.  Comparisons to measurements at hourly and daily time scales are shown in Figure 6 above for particle number concentration.  The hourly comparisons meet model performance criteria, but have slightly worse performance than the daily averages because the calculations do not fully capture all of the random variability in meteorological patterns and emissions patterns over hourly time scales.  Further work would be required to create accurate model results at hourly time scales, but this effort is beyond the reasonable scope of the current study.  We do not wish to present hourly-average performance metrics in the manuscript because we do not want to encourage the use of the model results at this time scale.

Unfortunately, number size distribution measurements are not available at the monitoring sites, only total number count.

Reviewer 2 Comment 3.Page 10,   equation 1: The method used to convert mass contribution to number contribution is questionable. First, which mass (mass size distribution or total mass) do you use in eq. (1)? It is not clear how you define the Dp. Second, the nucleation is a major source of particle number, but it won't contribute a lot to the mass concentration, so if you use the mass size distribution in eq. (1), then it is better to check the number size distribution of nucleation source to evaluate the method. Also, condensation is an important process for the growth of nucleation mode particles. So the change of density can not be ignore.

Response:  Please see the response to Review 1 Comment 1.1.  The number source apportionment methods have been extensively tested and verified as accurate.

Reviewer 2 Comment 4. Page 12, line 268: which method was used to measure PM0.1 in Xue's paper? In figures 2-3, the authors only compare the data in 2015 and 2016. I guess there was no measurement in 2012. But in Figures 7-9, you only show the time series of PM0.1 in 2012? This selective comparison is also shown for particle number concentration (figures 4-6). I would suggest the authors should also show the time series data in 2015 and 2016, which contain both measurement and modelling results. Additionally, what is the time resolution in Figures 4-9? It seems the x-axis in Figures 4 and 7 is not regular.

Response: The methods used to measure PM0.1 are explained in detail by Xue et al. (2018). Briefly, PM0.1 samples were collected on the final stage of a Micro Orifice Uniform Deposit Impactor (MOUDI). Samples were analyzed for chemical composition and then the Chemical Mass Balance (CMB) model was used to predict source contributions to PM0.1.

The focus of the current study is to demonstrate the model formulation and performance for the year 2012 but the comparison to measurements from 2015-16 is included to more fully confirm the model calculations are getting the contributions from each major source approximately correct. A future paper will discuss the time series results for the years 2000 through 2016 and so we would like to save the comparisons requested by the reviewer for that forthcoming paper.

The time resolution in manuscript Figures 4 through 7 is daily average for selected months. The x-axis in each of these figures has been updated to show the month more clearly in the revised manuscript.

Reviewer 2 Comment 5. In Figures 2-3, there is no nucleation source, if the authors use eq.1 to convert mass contribution to number contribution, then wow did the authors define "nucleation" source?

Response: Nucleation accounts for a significant amount of $N_7$ but a minor amount of $PM_{0.1}$ due to the very small size of these particles. This is shown in Figures 10-12 in the revised manuscript. Nucleated particles are tracked separately from emitted particles in the revised model calculations so that they can be quantified more exactly.

Reviewer 2 Comment 6. Nucleation is a major source of particle number concentration. I would suggest the authors also show the modelling results only for nucleation days. If you put it in the average data (figures 12-14), then more information might be covered. And why you only show the average data from August and December.

Response: Figures 4-6 in the revised manuscript show that some amount of nucleation occurs on most days across the monitoring sites included in the current study. Average diurnal variations are shown for August and December to capture the behavior in the hottest and coldest months of the year to span the range of possible temperature effects. This is clarified on line 563 of the revised manuscript.

Reviewer 3 Comments

Yu et al., use the UCD/CIT model to simulate ultrafine particulate matter in California, focusing on the Los Angeles and San Francisco areas. To do so, they have developed an inventory of relevant emissions and added a nucleation model to the code. They find acceptable model performance. A particular finding is that non-residential gas combustion is a dominant contributor.

Reviewer 3 Comment 1. The paper has some interesting aspects to it, particularly the assignment of sources to their impacts on particle number. This may also be its weakness as there is little means to assess the validity of some of the resulting conclusions that might be drawn and the results are striking and don't really line between the model simulations and the observations. Further, they don't bring in recent findings.

Response: The model predictions are in reasonable agreement with all $N_7$ and $PM_{0.1}$ measurements. The model predictions agree with the recent measurements of particle number plumes downwind of airports (see revised manuscript Figure 18 and associated discussion). The model predictions agree with measurements that show mobile sources dominate near highways but other sources become more important further away from highways (see revised manuscript Figure 4 and associated discussion). In summary, the results agree with measurements and recent findings published in the literature.

Reviewer 3 Comment 2. Their main result is that non-residential natural gas (NRNG) combustion is the major contributor to particle number often contributing over half. Looking at Fig. 10, NRNG contributes about 60-70% of the total at almost all the cities (slightly more at Rubidoux, somewhat less at Livermore). This is remarkably consistent given what has been found about the contribution of mobile sources and aircraft emissions to UFPs in other stud-ies (e.g., U Wash, USC studies). They don't include aircraft in these plots: this is a huge shortcoming, and on this alone, the manuscript requires much more work before being considered for publication. A major weakness here is also that the emissions from NRNG, vs. residential NG, is from a recently published manuscript. However, in my reading of that manuscript, they do not include the conditions referred to in this manuscript (a dilution factor of 25), and they seemed to focus on biogas. Maybe the use of the word "same" is of issue here as well. It should be noted that the observations also do not support that the main source is NRNG (and their model results suggest this as well), as particle number increases at night in December, starting about rush hour and going until about 8 pm. This very much looks like mobile source emissions, but cer-tainly not an industrially-related source that would likely decrease after 17:00. During the summer, there appears to be more of a mid-day, photochemically-generated peak. Overall, the observations tend to suggest something very different than the model.

Response: The revised manuscript Figure 4 shows that on-road mobile sources contribute significantly to particle number concentration near roadways and revised manuscript Figure 18 shows that aircraft contribute significantly to particle number concentrations downwind of major airports. The model results agree with the measurements for mobile sources and aircraft, but they go on to show that those measurements do not completely characterize the urban atmosphere. Natural gas combustion exhaust contributes significantly across the major regions where it is used heavily as a fuel.

The recent measurements of natural gas combustion exhaust performed by Xue et al. (2018) are in good agreement with the earlier measurements of natural gas combustion exhaust performed by Li and

Hopke (1993) as noted on line 658 of the revised manuscript. The dilution factor of 25 used by Xue et al. (2018) is noted on line 677 of the revised manuscript.

The diurnal contributions from natural gas combustion appear to agree with measurements at some locations but disagree with measurements at other locations as discussed for Figures 15 to 17 in the revised manuscript. We do not claim that model agreement with measurements is perfect, but we do not see evidence that rules out the importance of natural gas combustion as a significant source of ultrafine particles.

Reviewer 3 Comment 3. They make the statement that "traffic sources contributed to PNC but did not dominate over regions more than 300 m away from freeways." This is a rather strange state-ment given that their model resolution is 4 km. They have no way of supporting this statement. Their making this statement is worrisome.

Response: Please see revised manuscript Figure 4 and associated discussion. Model simulations were performed at scales ranging from 250m through 4km to illustrate this point more clearly.

Reviewer 3 Comment 4. They also state in the Abstract that the performance meets the threshold normally required for regulatory modeling. I am not aware that such a threshold has been set. I don't believe the Boylan and Russel paper is accepted by any agency. Further, they need to be much more informative as to how they actually calculated the performance statistics given that the number concentrations are available at a finer time resolution than the species concentrations often used in performance determinations. Maybe they should also look at the AQMEII studies. The current table of performance (Table 1) is insufficient.

Response: All references to regulatory modeling performance criteria have been removed from the manuscript as suggested by the reviewer. Comparisons in the manuscript are based on daily averages which corresponds to the shortest averaging time that should be used for the current model results. Comparisons to measurements at hourly and daily time scales are shown in Figure 6 above for particle number concentration. The hourly comparisons meet model performance criteria, but have slightly worse performance than the daily averages because the calculations do not fully capture all of the random variability in meteorological patterns and emissions patterns over hourly time scales. Further work would be required to create accurate model results at hourly time scales, but this effort is beyond the reasonable scope of the current study. We do not wish to present hourly-average performance metrics in the manuscript because we do not want to encourage the use of the model results at this time scale.

Table 1 has been revised to show observed particle number, simulated particle number, correlation coefficient, MFB, MFE, and RMSE.

Reviewer 3 Comment 5. Their reference to Shet et al., referring to Taylor's hypothesis, is not relevant here. Tay-lor was looking at turbulence correlations, and the relationship between temporal fluctu-ations and spatial fluctuations. Here, one has to assume that emissions and chemistry play a huge part, particularly since the observations are averaged over scales much larger than the Taylor scale. It was not even apparent why they cited the paper.

Response: The reference to Taylor's hypothesis was included based on a previous review of an early version of the manuscript. We would be happy to remove it but defer to the Editor's judgement.

Reviewer 3 Comment 6. Looking at Fig. 12, there are a number of locations where there appears to be a mismatch between 23:00 and 0:00.

Response: Figures 15 to 17 in the revised manuscript use hours 1 through 24 on the lower axis. These are separate hours in the model simulation driven by inventories that do not necessarily have matching emissions at hour 1 and 24, and so some discontinuity is expected.

Reviewer 3 Comment 7. Boundary conditions can be very important in regions close to the coast. A diagram of the modeling domain should be provided along with the boundary conditions. A test of the impact of boundary conditions on the results should be provided.

Response: The modeling domains are illustrated in Figure 2 of the revised manuscript. The source-tagging features inherent in the model allow us to evaluate the influence of boundary conditions without resorting to a brute force sensitivity study. The influence of major sources of particle number are shown in Figure 19 of the revised manuscript. Boundary conditions make negligible contributions and thus are not shown.

Reviewer 3 Comment 8. Their modeling domain height is only 5 km. This is lower than most any other model used, from what I recall. Citing some of their studies without really doing a comparison as to the impact of having a higher domain is not sufficient.

Response: WRF/Chem simulations conducted with a model height greater than 12km clearly show that the vast majority of the urban pollution that affects ground level concentrations is contained within the first few km of the atmosphere. We will be happy to show results in the current manuscript if the Editor believes this is a good use of space, but we respectfully suggest that it is not necessary to test every basic feature of the modeling system for California that has been previously evaluated in published results.

Reviewer 3 Comment 9. What is meant by "Model source code and model input files are available to collabo-rators via direct email. . . It should be made available to anyone looking to check their results. A more general statement of availability should be provided. All files and data needed to recreate the results should be available.

Response: All emissions inputs, spatial surrogates, field measurements used to generate model inputs and evaluate model results were obtained from air pollution control agencies who will make these same datasets available to any modeling groups wishing to re-create the inputs. The size and composition distribution profiles used to generate emissions of ultrafine particulate matter are publicly available in peer-reviewed journal articles published over the past several decades. Once again, these data are available to anyone wishing to re-create the inputs. If fellow researchers do not wish to go to the trouble of accessing the publicly available information to assemble the model inputs for themselves, then the authors are also willing to collaborate as stated in the manuscript. Please contact the corresponding author to discuss future collaborations.

Reviewer 3 Comment 10. Fig. 3. Two issues here. First, the caption suggests that both CMB and UCD results are shown. Are CMB results labeled as "Obs.". This would be a wrong interpreta-tion. Further, how are the uncertainties determined? Second, they should also show secondary fractions.

Response: Yes, CMB results are labelled as "Obs" since this is a simpler model driven by direct measurements. Clearly one cannot observe source contributions – they are inferred by some statistical receptor model such as CMB or PMF. We believe this is a common understanding in the field.

Uncertainties are determined using the standard CMB error estimation techniques.

Secondary fractions are lumped into the "others" category in CMB calculations and hence are already shown on the Figure.

Reviewer 3 Comment 11. Figs. 5-6, a correlation plat would be useful. The obs seem to be rather less variable.

Response: Pearson correlation coefficients for each site are now shown in revised Table 1.

Reviewer 3 Comment 12. Fig. 10-11. These results bring up a question: Were the same size distribution on the emissions used everywhere on a source-by-source basis.

Response: Each emissions size distributions for each source is constant at all locations. Emitted particles evolve to different size distributions at different locations through atmospheric aging processes that vary by location.

Reviewer 3 Comment 13. Line 514: It should be "under".

Response: Sentence revised.

Reviewer 3 Comment 14. Summary: At present, there a serious issues with the paper, including not including aircraft impacts, that the result that NRNG is the dominant contributor does not appear to explain the observations, some statements that are off-base (scale of impact of freeways, performance metric for regulatory acceptance, Taylor's hypothesis) and the need to better describe how performance was evaluated. A major rewrite, alone, may not be able to address all of the concerns.

Response: Aircraft impacts are included in the revised manuscript. On-road mobile sources as a function of spatial resolution are quantified in the revised manuscript. The role of natural gas combustion exhaust is put into better perspective relative to past measurement studies in the revised manuscript. References to regulatory performance metrics are removed in the revised manuscript.

We respectfully feel that this paper has been rigorously updated to meet all reasonable comments.

n̶a̶t̶u̶r̶a̶l̶ ̶g̶a̶s̶ ̶c̶o̶m̶b̶u̶s̶t̶i̶o̶n̶ ̶(̶4̶2̶-5̶7̶%̶)̶ ̶w̶a̶s̶ ̶t̶h̶e̶ ̶l̶a̶r̶g̶e̶s̶t̶ ̶P̶M̶$_{0.1}$ ̶s̶o̶u̶r̶c̶e̶ ̶a̶t̶ ̶t̶h̶e̶ ̶S̶o̶C̶A̶B̶ ̶s̶i̶t̶e̶s̶,̶
35   f̶o̶l̶l̶o̶w̶e̶d̶ ̶b̶y̶ ̶t̶r̶a̶f̶f̶i̶c̶ ̶s̶o̶u̶r̶c̶e̶s̶ ̶(̶1̶6̶-3̶5̶%̶)̶ ̶a̶n̶d̶ ̶f̶o̶o̶d̶ ̶c̶o̶o̶k̶i̶n̶g̶ ̶(̶6̶-1̶4̶%̶)̶.̶ The study region
encompassed in this project is home to more than 25M residents, which should provide
sufficient power for future epidemiological studies on the health effects of airborne
ultrafine particles. C̶o̶r̶r̶e̶l̶a̶t̶i̶o̶n̶s̶ ̶b̶e̶t̶w̶e̶e̶n̶ ̶P̶M̶$_{2.5}$ ̶a̶n̶d̶ ̶P̶N̶C̶ ̶a̶r̶e̶ ̶l̶o̶w̶ ̶(̶R̶$^2$=̶0̶.̶3̶5̶)̶ ̶s̶u̶g̶g̶e̶s̶t̶i̶n̶g̶
t̶h̶a̶t̶ ̶t̶h̶e̶ ̶h̶e̶a̶l̶t̶h̶ ̶e̶f̶f̶e̶c̶t̶s̶ ̶o̶f̶ ̶t̶h̶e̶s̶e̶ ̶m̶e̶t̶r̶i̶c̶s̶ ̶m̶a̶y̶ ̶b̶e̶ ̶a̶s̶s̶e̶s̶s̶e̶d̶ ̶i̶n̶d̶e̶p̶e̶n̶d̶e̶n̶t̶l̶y̶.̶ All of the PM$_{0.1}$

[revised manuscript text omitted]

---

## Author Response (AR3)

Reviewer 2 Comments

The simulations of particle number concentration and the corresponding source apportionment are very difficult in the model. We should encourage the relevant research. Still, more work should be included in the further study, especially for the model verification.

Comment 1. Maybe it is reasonable to set the first size bin as 7 nm in the model calculation, which will benefit the comparison.

Response: The reviewer makes a good suggestion to remove the uncertainty associated with comparing N7 measurements to N10 model predictions.  We will incorporate this suggestion into future studies.

2. Page 12, line 289: I do not understand the Table 2. Particle number concentration and PM2.5 came from the different sources, so why you demonstrate the correlation ($R^2$) ? Also, you use R in table 1, and $R^2$ in table 2, why?

Response: The analysis showing correlation coefficients between particle number and PM2.5 was requested by a previous reviewer as another metric to test model performance.  Predicted correlation coefficients tend to be higher than measured correlation coefficients suggesting that the model calculations do not capture all of the complexity in the real atmosphere.  This has been clarified on line 294 of the revised manuscript.  As noted in the current text, some of the higher correlation between particle number and PM2.5 in the model results may be caused by the artificial smoothing inherent in the 4km model grid cells.

The correlation coefficient in Table 2 has been changed to R to match with Table 1.

3. What is the Y-axis label for Figs. 10-12?

Response: The y-axis label should read $PM_{0.1}$ ($\mu$g m$^{-3}$).  This has been corrected in the revised manuscript.

4. I think the figure label is wrong in Fig. 15, also why the scale of the x-axis in Figs. 15-17 is from 1 to 24?

Response: The figure label in Fig 15 has been corrected to label the panels a,b,c,d,e,f,g,h.  The x-axis scale representing hours of the day as part of the diurnal profile has been revised to properly show hours 1-24 counting by 2.

5. There are several grammar mistakes in the text, the language and symbols should be checked carefully once more before publication.

Response: The text has been carefully proofread and necessary corrections have been made.

[revised manuscript text omitted]

---

## Author Response (AR4)

Editor Comments

Comment 1. Line 152: please include some more recent publications on this, such as doi:10.5194/acp-15-13993-2015, doi:10.1126/science.aaf2649, doi.org/10.5194/acp-19-8591-2019, etc.

Response: References cited as requested.